# M²DDI: A Unified Framework for Dynamic Multimodal Fusion in Drug-Drug Interaction Prediction

## Abstract

Drug-drug interaction (DDI) prediction is critical for ensuring patient safety and optimizing therapeutic outcomes. Existing computational approaches are limited by their inability to jointly model the heterogeneous mechanisms underlying DDIs, which span molecular structure, pharmacodynamic function, and network-mediated relations. To address this limitation, we introduce `M²DDI`, a unified framework for dynamic multimodal fusion in DDI prediction. `M²DDI` utilizes a Mixture-of-Experts architecture, with each expert dedicated to a distinct pharmacological modality. A novel prior-enhanced dual-path gating strategy adaptively selects relevant experts for each drug pair by integrating mechanism-matched feature queries and ATC-based biomedical priors, thereby aligning expert selection with underlying pharmacological mechanisms and addressing the challenge of data incompleteness. Empirical evaluation on benchmark datasets demonstrates that `M²DDI` achieves state-of-the-art performance, particularly in new drug scenarios. Additional robustness experiments show that `M²DDI` maintains high predictive accuracy even when modality-specific information is partially missing, outperforming existing methods under similar conditions. Analysis of expert selection patterns further confirms alignment with established pharmacological mechanisms. These results establish `M²DDI` as an effective and mechanism-aware solution for comprehensive DDI prediction. The code are available at https://anonymous.4open.science/r/M2DDI-AECB

## 1 Introduction

A drug-drug interaction (DDI) occurs when the effect of one drug is altered by the co-administration of another, potentially leading to adverse reactions or reduced therapeutic efficacy (Magro et al., 2012). Therefore, accurately forecasting these interactions is paramount for maximizing therapeutic benefit while minimizing patient harm (Roemer & Boone, 2013; Su et al., 2022).

The mechanisms underlying drug-drug interactions (DDIs) are heterogeneous, comprising pharmacokinetic (PK), pharmacodynamic (PD), and relational mechanisms(Palleria et al., 2013). PK mechanisms (structural) often involve structural similarities that affect absorption, distribution, metabolism, or excretion between drugs; existing methods employ molecular graph representations to model these mechanistic features (Rogers & Hahn, 2010; Ryu et al., 2018). PD mechanisms (functional) arise when drugs act on overlapping biological targets; structured textual information from drug databases is leveraged by current approaches to represent these relationships (Zhu et al., 2023; Abdullahi et al., 2025). Relational mechanisms (relational) reflect indirect interactions through complex pathways among drugs and other biomedical entities; knowledge graph (KG)-based approaches are designed to capture these dependencies (Zitnik et al., 2018; Yu et al., 2021; Zhang et al., 2023).

Given the complex mechanism of DDIs, the different modalities, including molecular structures, textual descriptions and KG triplets, indicate the potential interactions in distinct aspects. As shown in the Table 1, existing methods mainly focus on single modality, which is insufficient for fully understand the mechanism of complex DDIs. In addition, real-world DDI resources are notoriously *incomplete and imbalanced* across modalities. This leads to systematic *modality incompleteness* and *modality imbalance* at the drug and drug-pair level, where some modalities are missing or severely

under-informative. Therefore, a unified framework that appropriately fuses different modalities is essential for comprehensive DDI prediction.

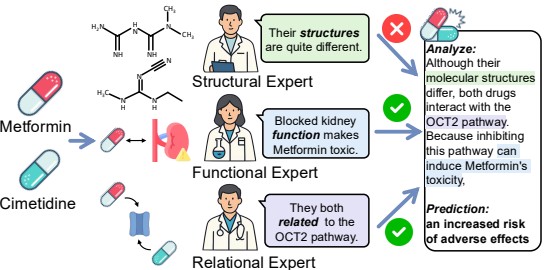

Figure 1: An illustration of modality heterogeneity and dynamic relevance in DDI task.

Table 1: Comparison of baselines by modality coverage and fusion strategy

| Method | Struct. | Func. | Rel. | Fusion |
|---|---|---|---|---|
| MLP (Rogers & Hahn, 2010) | ✓ | ✗ | ✗ | – |
| DeepDDI (Ryu et al., 2018) | ✓ | ✗ | ✗ | – |
| TextDDI (Zhu et al., 2023) | ✗ | ✓ | ✗ | – |
| Decagon (Zitnik et al., 2018) | ✗ | ✗ | ✓ | – |
| SumGNN (Yu et al., 2021) | ✗ | ✗ | ✓ | – |
| EmerGNN (Zhang et al., 2023) | ✗ | ✗ | ✓ | – |
| TIGER (Su et al., 2024) | ✓ | ✗ | ✓ | Static |
| **M²DDI (Ours)** | ✓ | ✓ | ✓ | **Dynamic** |

A naive solution is to statically combine modalities through concatenation or ensembling (Li & Tang, 2024); however, as shown in Figure 1, this strategy fails to address three fundamental challenges in DDI prediction. First, **Modality Heterogeneity**. The modalities exhibit substantial differences in both structure and semantics (Abbas et al., 2025), rendering simple concatenation insufficient. Second, **Dynamic Relevance**. DDI mechanisms are heterogeneous: PK-driven interactions are typically dominated by molecular structure, whereas PD-driven ones rely more on functional targets (Yu et al., 2024). Static fusion ignores this distinction, often introducing noise from irrelevant modalities rather than attending to the critical signal. Third, **Data Incompleteness**. Real-world deployment faces an inherent *modality asymmetry* (Gangwal et al., 2024): while molecular structures are always available for novel drugs, their relational connections or textual descriptions are often missing (cold-start). Static fusion models often fail catastrophically when such auxiliary modalities are absent, limiting their generalization to new drugs. These challenges underscore the need for a framework capable of dynamically and selectively weighting the most pertinent evidence for each drug pair.

Building on this insight, we propose **M²DDI** (**M**ultimodal **M**ixture-of-Experts **D**rug-**D**rug **I**nteraction prediction), a unified framework that dynamically fuses multiple modalities for comprehensive DDI prediction. Inspired by advances in multimodal fusion (Li et al., 2025), M²DDI adopts a **Mixture-of-Experts** (MoE) architecture, with specialized expert models dedicated to distinct modalities, enabling targeted modeling of heterogeneous modality-specific information. To address the unique complexities of DDI mechanisms, M²DDI introduces a novel prior-enhanced dual-path gating strategy specifically designed for this task. A feature-query path employs mechanism-matched feature queries to dynamically assess the relevance of each modality for drug pairs, enhancing the modeling of mechanism heterogeneity. Another one, called prior path, incorporates biomedical knowledge priors to guide expert selection when data are incomplete or missing. Enhanced with the dual prior paths, the gating network ensures expert selection is both pharmacologically informed and resilient to data sparsity. The main contributions are summarized as follows:

- **A Unified Multimodal Fusion Framework:** M²DDI dynamically integrates molecular structure, textual, and knowledge graph modalities to comprehensively model heterogeneous pharmacological mechanisms in DDIs.

- **Prior-Enhanced Dual-Path Gating:** M²DDI employs a Mixture-of-Experts architecture that includes a novel dual-path gating strategy, combining mechanism-matched feature queries and biomedical priors for dynamic expert selection, directly addressing mechanism heterogeneity and data incompleteness.

- M²DDI achieves state-of-the-art performance across multiple benchmarks, exhibiting superior generalization to novel drugs and robust performance under data incompleteness, and further analysis reveals that its expert selections align with established pharmacological mechanisms in real-world DDI cases.

## 2 RELATED WORK

### 2.1 DRUG-DRUG INTERACTION PREDICTION

Computational drug-drug interaction (DDI) prediction has progressed through three primary paradigms, each aligned with a distinct pharmacological mechanism: structure-centric (pharmacokinetic), function-centric (pharmacodynamic), and relation-centric (relational). Each paradigm leverages a specific data modality, resulting in fragmented modeling of DDI mechanisms.

**Structure-Centric Methods** Structure-centric methods focus on pharmacokinetic mechanisms by modeling molecular structures to predict DDIs. Early approaches utilized machine learning on hand-crafted molecular descriptors (Rogers & Hahn, 2010; Cheng & Zhao, 2014; Zakharov et al., 2016), while subsequent works adopted deep learning on SMILES strings or molecular graphs to automate feature extraction (Ryu et al., 2018; Hou et al., 2019). These methods effectively capture absorption, distribution, metabolism, and excretion properties. However, they are inherently limited to structural information and cannot represent pharmacodynamic or relational mechanisms.

**Function-Centric Methods** Function-centric methods address pharmacodynamic mechanisms by extracting information from structured textual sources, such as drug indications and mechanisms of action (Zhu et al., 2023; Liu et al., 2025). Natural language processing techniques are employed to model functional relationships between drugs. The effectiveness of these methods depends on the availability and quality of textual data, which may be sparse or inconsistent for novel or less-studied drugs.

**Relation-Centric Methods** Relation-centric methods target relational mechanisms by modeling DDIs as link prediction or reasoning tasks on biomedical knowledge graphs (KGs) (Himmelstein & Baranzini, 2015; Chandak et al., 2023; Abdullahi et al., 2025). Drugs are represented as nodes, and interactions are inferred through graph-based algorithms, including graph neural networks (Zitnik et al., 2018; Zhang et al., 2023) and graph transformers (Su et al., 2024). The performance of these methods is constrained by the completeness and coverage of the underlying KG. Many drugs in benchmark datasets are either absent or isolated in standard KGs (Walsh et al., 2020), limiting the applicability of relation-centric approaches.

In summary, existing DDI prediction methods are segregated by modality and mechanism, resulting in incomplete modeling of the heterogeneous nature of DDIs. $\texttt{M}^2\texttt{DDI}$ addresses this limitation by providing a unified framework for comprehensive multimodal fusion in DDI prediction.

### 2.2 MULTIMODAL FUSION

Multimodal fusion seeks to integrate heterogeneous data sources to improve predictive modeling in tasks (Baltrušaitis et al., 2018). Conventional fusion methods, such as static fusion and ensemble approaches (Li & Tang, 2024), typically concatenate features from each modality and process them with standard neural architectures (Liu et al., 2018; Tsai et al., 2019; Xue et al., 2023), but such approaches do not account for the diverse and context-dependent relationships among modalities. The Mixture-of-Experts (MoE) architecture enables modeling of interactions through expert specialization (Jacobs et al., 1991; Chen et al., 1999; Yuksel et al., 2012). Recent studies (Mustafa et al., 2022; Yu et al., 2023) havie investigated the application of MoE in multimodal learning contexts. However, the dominant paradigm applies MoE to a shared, pre-fused feature space (Li et al., 2025), implicitly assuming the consistent relevance of all modalities. This premise is ill-suited for DDI prediction, where heterogeneous mechanisms mean a single modality may be critical while others are noisy or irrelevant. Fusing them indiscriminately can dilute the primary signal. This motivates an alternative framework where experts are architecturally bound to specific, mechanism-aligned modalities, shifting the task from implicit pattern specialization to explicit evidence source selection.

Recently, methods like MOF-DDI (Wen et al., 2023) have introduced a *Static Alignment* paradigm using Optimal Transport to unify heterogeneous embeddings. However, forcing alignment implies a multimodal consensus, which conflicts with the *Mechanism Heterogeneity* of DDIs—often driven by a single dominant factor rather than all modalities—and fails under *Modality Incompleteness* when alignment targets are missing for novel drugs. In contrast, our framework shifts to *Dynamic*

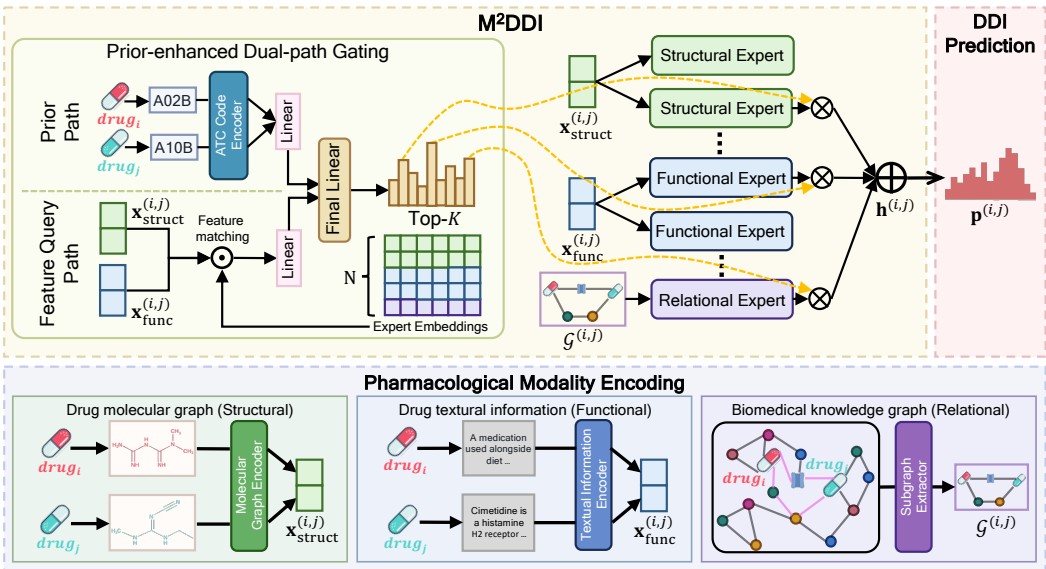

Figure 2: Overview of the M²DDI framework, illustrated with the query: *"what's the interaction between Metformin and Cimetidine?"*. The model processes multi-modal inputs (structural, functional, relational) for a drug pair. A prior-enhanced dual-path gating mechanism then activates the top-$K$ most relevant experts from a heterogeneous pool, and their outputs are combined via a weighted sum to produce the final interaction prediction.

*Routing*, treating modalities as independent experts to actively select the optimal evidence source, ensuring robustness even with partial data.

## 3 METHODOLOGY

### 3.1 OVERALL FRAMEWORK

Given a drug pair $(drug_i, drug_j)$, the objective is to predict the interaction type from a predefined set of types $\mathcal{T}_{\texttt{DDI}}$. This task is addressed using the proposed M²DDI framework. As illustrated in Figure 2, the framework integrates several core modules. The process begins with **Pharmacological Modality Encoding**, wherein each drug's structural and functional facets are converted into feature representations, while its relational context is captured as a distinct subgraph. These multi-modal inputs are then processed by our **M²DDI architecture**, which features a heterogeneous expert pool and a novel **prior-enhanced dual-path gating strategy**. The outputs from the selected experts are subsequently integrated for predicting the target interaction types.

### 3.2 PHARMACOLOGICAL MODALITY ENCODING

Each drug pair is represented by three modalities. The structural representation, $\mathbf{x}_{\texttt{struct}}^{(i,j)} \in \mathbb{R}^{2d_{\texttt{embed}}}$, is obtained by separately encoding the molecular graphs of $drug_i$ and $drug_j$ with a molecular graph encoder, then concatenating the outputs. The functional representation, $\mathbf{x}_{\texttt{func}}^{(i,j)} \in \mathbb{R}^{2d_{\texttt{embed}}}$, is derived by separately encoding textual information for each drug using a textual information encoder, followed by concatenation. The relational modality, $\mathcal{G}^{(i,j)}$, is constructed as a subgraph containing all directed paths of length at most $L$ from $drug_i$ to $drug_j$ in a biomedical knowledge graph. These encoded outputs are used for subsequent fusion. Further details are provided in Appendix C.

### 3.3 M²DDI: UNIFIED MULTIMODAL FUSION ARCHITECTURE

With the drug pair's structural and functional representations established and its relational subgraph formulated, we now introduce the core of our reasoning engine: the M²DDI architecture. This

architecture is designed to dynamically fuse these multimodal inputs through the synergy of two key components: a **heterogeneous expert pool** as the primary analysis unit and a **prior-enhanced dual-path gating strategy** for expert selection.

### 3.3.1 HETEROGENEOUS EXPERT POOL

The $\texttt{M}^2\texttt{DDI}$ framework employs a unified pool of $N = N_s + N_f + N_r$ heterogeneous experts, each assigned to a specific pharmacological modality. **Structural experts** ($\{E_{\texttt{struct}}^{(n)}\}_{n=1}^{N_s}$) process the input $\mathbf{x}_{\texttt{struct}}^{(i,j)}$ to model pharmacokinetic mechanisms. **Functional experts** ($\{E_{\texttt{func}}^{(n)}\}_{n=1}^{N_f}$) analyze $\mathbf{x}_{\texttt{func}}^{(i,j)}$ to capture pharmacodynamic mechanisms. **Relational experts** ($\{E_{\texttt{rel}}^{(n)}\}_{n=1}^{N_r}$) operate on the subgraph $\mathcal{G}^{(i,j)}$ to represent relational pathways. For efficient computation, both structural and functional experts are implemented as multilayer perceptrons, while relational experts utilize graph neural networks to capture complex dependencies within the biomedical knowledge graph. Each expert $E_n$ produces a hidden representation $\mathbf{h}_n^{(i,j)} = E_n(\texttt{input}^{(i,j)}) \in \mathbb{R}^{d_{\texttt{hidden}}}$, which is used for subsequent multimodal fusion and prediction.

### 3.3.2 PRIOR-ENHANCED DUAL-PATH GATING STRATEGY

The prior-enhanced dual-path gating strategy is designed to dynamically select modality-specific experts for each drug pair by integrating mechanism-matched feature queries and pharmacological knowledge priors. This approach addresses the limitations of conventional gating mechanisms, which typically rely on a single linear transformation of input features and cannot adequately capture the heterogeneous mechanisms underlying drug-drug interactions.

**The first path**, referred to as the feature-query path, computes expert relevance scores by matching the concatenated structural and functional representations of the drug pair with a set of learnable expert embeddings. Formally, given the drug-pair feature vector $\texttt{Concat}(\mathbf{x}_{\texttt{struct}}^{(i,j)}, \mathbf{x}_{\texttt{func}}^{(i,j)})$, the relevance score for each expert is calculated via a dot product with the expert embedding matrix $\mathbf{E} \in \mathbb{R}^{N \times d_{\texttt{query}}}$, followed by a learnable transformation:

$$\mathbf{s}_{\texttt{feat}} = (\texttt{Concat}(\mathbf{x}_{\texttt{struct}}^{(i,j)}, \mathbf{x}_{\texttt{func}}^{(i,j)})\mathbf{E}^\top)\mathbf{W}_{\texttt{feat}}, \tag{1}$$

where $\mathbf{W}_{\texttt{feat}} \in \mathbb{R}^{N \times N}$ models inter-expert relationships. This design enables the gating mechanism to align drug-pair features with the modality-specific analysis capabilities of each expert, rather than relying on a single linear projection of the input.

**The second path**, referred to as the prior path, incorporates pharmacological knowledge from the Anatomical Therapeutic Chemical (ATC) classification system. The ATC system is a hierarchical drug classification scheme that encodes pharmacological, therapeutic, and chemical properties of drugs. Each drug is assigned one or more ATC codes, which are structured into multiple levels: for example, the code "A10B" can be decomposed into three levels, $l_1 =$"A", $l_2 =$"A10", and $l_3 =$"A10B". For each drug, the ATC-based representation is constructed by aggregating the hierarchical embeddings of its assigned ATC codes. Specifically, each code's feature $\mathbf{z}_c$ is generated by concatenating its level embeddings and projecting them into the ATC space:

$$\mathbf{z}_c = \texttt{Concat}(\mathbf{e}_{11}, \mathbf{e}_{12}, \mathbf{e}_{13})\mathbf{W}_{\texttt{ATC-embed}}, \tag{2}$$

$$\mathbf{a}^{(k)} = \texttt{MeanPooling}_{c \in \mathcal{C}_k}(\mathbf{z}_c), \tag{3}$$

where $\mathcal{C}_k$ denotes the set of ATC codes assigned to drug $k$, and $\mathbf{W}_{\texttt{ATC-embed}} \in \mathbb{R}^{d_{\texttt{hierarchical}} \times d_{\texttt{ATC}}}$ is a learnable projection matrix. The ATC-based representations of the drug pair are then concatenated and transformed to yield expert relevance scores:

$$\mathbf{s}_{\texttt{ATC}} = \texttt{Concat}(\mathbf{a}^{(i)}, \mathbf{a}^{(j)})\mathbf{W}_{\texttt{ATC-score}}, \tag{4}$$

where $\mathbf{W}_{\texttt{ATC-score}} \in \mathbb{R}^{(2d_{\texttt{ATC}}) \times N}$ is a learnable matrix that projects the concatenated ATC features to expert scores. To robustly handle drugs with unavailable biomedical priors, we assign a learnable 'missing embedding' parameter $e_{missing} \in \mathbb{R}^{d_{ATC}}$ to represent the absence of ATC codes. This design ensures that the Prior Path remains fully operational even when knowledge data is incomplete, allowing the gating mechanism to adaptively fall back to the Feature-Query path ($\mathbf{s}_{\texttt{feat}}$) by relying on the available structural or functional evidence (see robustness analysis in Appendix M). This

path introduces external pharmacological priors into the gating process, providing complementary evidence to the feature-query path, particularly in cases of incomplete or sparse modality-specific features. Crucially, this design is intended to mitigate the challenge of data incompleteness; since ATC classification is a standard for nearly all drugs, it provides a reliable signal for the gating mechanism when other modalities, particularly for novel compounds, are sparse or absent.

The scores from the feature-query path ($\mathbf{s}_{\text{feat}}$) and the prior path ($\mathbf{s}_{\text{ATC}}$) are independently computed for all experts and then concatenated. The final expert gating scores are obtained by integrating these dual-path scores through a routing matrix and applying a sigmoid activation:

$$\mathbf{g} = \sigma(\texttt{Concat}(\mathbf{s}_{\text{feat}}, \mathbf{s}_{\text{ATC}})\mathbf{W}_{\texttt{gate}}) \in \mathbb{R}^N. \tag{5}$$

This dual-path strategy ensures that both mechanism-matched feature queries and pharmacological priors jointly inform expert selection, enabling adaptive and mechanism-aware multimodal fusion for drug-drug interaction prediction.

### 3.4 Prediction and Training

This part details the final steps: ensuring balanced expert utilization, synthesizing expert outputs for prediction, and the overall training objective.

**Loss-Free Load Balancing.** Given the significant heterogeneity of our expert pool (e.g., MLPs vs. GNNs), traditional auxiliary loss (Lepikhin et al., 2020) for load balancing is suboptimal. We therefore adopt a loss-free strategy (Wang et al., 2024) to ensure balanced expert utilization without interfering with the task gradient. A learnable, input-independent bias vector, $\mathbf{b} \in \mathbb{R}^N$, perturbs the gating scores $\mathbf{g}$ before selecting the top-$K$ experts, whose indices are denoted by $\mathcal{I}$:

$$\mathcal{I} = \texttt{argtopk}_{n=1}^{N}(\mathbf{g} + \mathbf{b}). \tag{6}$$

The bias vector $\mathbf{b}$ is updated separately from the task gradient to counteract routing imbalance, guided by the expert utilization fraction, $\mathbf{F} \in \mathbb{R}^N$, in each batch:

$$\mathbf{b} \leftarrow \mathbf{b} - \alpha \cdot \texttt{sign}(\mathbf{F} - \frac{1}{N}), \tag{7}$$

where $\alpha$ is a hyperparameter controlling the update step size. Through this mechanism, the scores of over-utilized experts are suppressed, while those of under-utilized experts are amplified, thereby dynamically encouraging a uniform distribution of expert selection.

**Final Prediction.** Once the gate identifies the top-$K$ experts, indexed by $\mathcal{I}$, their outputs are synthesized. The final unified representation, $\mathbf{h}^{(i,j)}$, is computed as a weighted sum of the selected experts' outputs $\{\mathbf{h}_n^{(i,j)}\}_{n \in \mathcal{I}}$. The contribution weights are determined by applying the softmax function to the raw gating scores $\{g_n\}_{n \in \mathcal{I}}$ of the selected experts, which were generated by the gating mechanism as defined in equation 5:

$$\mathbf{h}^{(i,j)} = \sum_{n \in \mathcal{I}} \frac{\exp(g_n)}{\sum_{m \in \mathcal{I}} \exp(g_m)} \cdot \mathbf{h}_n^{(i,j)}. \tag{8}$$

The resulting representation $\mathbf{h}^{(i,j)} \in \mathbb{R}^{d_{\texttt{hidden}}}$ is subsequently passed to a final classifier, which is implemented as a multilayer perceptron, to produce the interaction to produce the interaction logits:

$$\mathbf{p}^{(i,j)} = \texttt{Classifier}(\mathbf{h}^{(i,j)}) \in \mathbb{R}^{|\mathcal{T}_{\texttt{DDI}}|}. \tag{9}$$

**Training Objective.** The entire $\texttt{M}^2\texttt{DDI}$ framework is trained end-to-end by minimizing the cross-entropy loss. The objective is to optimize the model parameters such that the predicted logits $\mathbf{p}^{(i,j)}$ for a given drug pair align with the ground-truth interaction type. We detail the specific loss formulation in Appendix D and provide analyses of computational complexity and empirical efficiency in Appendix I and J, respectively.

## 4 EXPERIMENTS

### 4.1 EXPERIMENTAL SETUP

**Experimental Setup.** We conduct 5-fold cross-validation experiments on two widely used DDI datasets: **DrugBank** (Wishart et al., 2018) (a multi-class classification task with 86 interaction types) and **TWOSIDES** (Tatonetti et al., 2012) (a multi-label classification task 200 side effects), with further details in Appendix F.1. Following established benchmarks (Dewulf et al., 2021; Zhang et al., 2023; Abdullahi et al., 2025), we evaluate across three settings designed to systematically assess the model's ability to generalize from familiar to entirely new pharmacological entities: **S0** (transductive, both drugs are seen in training), **S1** (inductive, with one new drug), and **S2** (fully inductive, with drugs unseen in training). To prevent data leakage, all textual information and knowledge graph data are strictly partitioned to ensure that no information from the test set is accessible during training. For DrugBank, we report macro-averaged F1-Score (primary) and Accuracy. For TWOSIDES, we report PR-AUC (primary) and ROC-AUC. Definitions for all metrics are provided in Appendix F.2.

**Implementation Details.** We implement the Molecular Graph Encoder using GraphMVP (Liu et al., 2021) framework to process molecular graphs derived from SMILES strings, and the Textual Information Encoder using BioMedBERT (Chakraborty et al., 2020) to process concatenated textual fields (name, description, pharmacodynamics, and mechanism-of-action) from DrugBank. To align modalities, the encoders and their projection heads are components pre-trained with the MoleculeSTM framework (Liu et al., 2023), and their parameters are kept frozen during task-specific training. For the Relational Expert's GNN module, we adopt the configuration from Zhang et al. (2023) and use HetioNet (Himmelstein & Baranzini, 2015) as the biomedical KG, setting the subgraph extraction length to $L = 3$. Our structural and functional experts are implemented as two-layer MLPs with ReLU activation and dropout. We configure a routable pool of nine experts in total ($N_s = 4$ structural, $N_f = 4$ functional, and $N_r = 1$ relational), activating the top $K = 5$ experts for each prediction via a final MLP head that shares the same two-layer architecture. Further details are provided in the Appendix F.4.

**Baseline Methods.** We compare $M^2DDI$ against a comprehensive suite of baselines categorized by data modality. Structure-centric methods include MLP (Rogers & Hahn, 2010) and DeepDDI (Ryu et al., 2018). For the function-centric methods, we use TextDDI (Zhu et al., 2023). The relation-centric baselines include Decagon (Zitnik et al., 2018), SumGNN (Yu et al., 2021), EmerGNN (Zhang et al., 2023), and TIGER (Su et al., 2024) (see Appendix F.3 for a detailed introduction). All of these baselines use one or at most two modalities, as shown in Table 1.

Finally, to specifically validate $M^2DDI$'s dynamic gating, we implement two non-adaptive multimodal fusion baselines: **Static Fusion** (Early Fusion), where expert embeddings are concatenated and fed to a unified MLP classifier; and **Ensemble** (Late Fusion), where each expert first makes independent predictions (logits), which are then concatenated and processed by a final linear layer for the ultimate output.

### 4.2 PERFORMANCE COMPARISON

The comprehensive evaluation results are shown in Table 2. $M^2DDI$ achieves significant gains (verified by the small $p$ values) over state-of-the-art baselines in DDI prediction. We provide the detailed observations and analysis as follows.

Unimodal baselines expose significant modality-specific limitations. Structure-centric models, such as DeepDDI, exhibit poor generalization, as evidenced by an F1 score drop from 73.55 to 13.73 on DrugBank (S0 vs. S2). Although function-centric approaches offer improved generalization, they are still inadequate. Relation-centric models like EmerGNN, while achieving strong inductive performance as the runner-up in DrugBank S1, are ultimately outperformed. These results collectively indicate that no single modality-specific paradigm consistently achieves robust performance.

Static, non-adaptive fusion strategies also demonstrate limited reliability. For example, the static fusion baseline consistently underperforms the best unimodal expert in all TWOSIDES settings, indicating that a naive combination of features can systematically degrade performance. Even more advanced approaches, such as the Stacking Ensemble, fail to close this gap. These results underscore

Table 2: Performance comparison, with best results **boldfaced** and second-best underlined. Asterisks denote statistical significance of M²DDI against the runner-up model (two-sided t-test): $^*p < .05$, $^{**}p < .01$. Full table with standard deviations in Appendix G.

| Category | Method | DrugBank | | | | | | TWOSIDES | | | | | |
| | | S0 | | S1 | | S2 | | S0 | | S1 | | S2 | |
| | | F1 | Acc | F1 | Acc | F1 | Acc | PR-AUC | ROC-AUC | PR-AUC | ROC-AUC | PR-AUC | ROC-AUC |
|---|---|---|---|---|---|---|---|---|---|---|---|---|---|
| Structure | MLP | 60.75 | 82.57 | 21.52 | 46.54 | 20.15 | 39.85 | 81.71 | 81.82 | 81.70 | 81.61 | 64.84 | 58.52 |
| | DeepDDI | 73.55 | 88.26 | 34.36 | 36.31 | 13.73 | 15.86 | 89.69 | 91.85 | 83.06 | 83.76 | 68.51 | 62.13 |
| Function | TextDDI | 92.00 | 95.75 | 59.51 | 66.40 | 26.79 | 44.23 | 92.67 | 94.56 | 84.29 | 85.83 | 83.18 | 78.16 |
| Relation | Decagon | 56.91 | 86.97 | 24.66 | 47.63 | 6.12 | 22.47 | 90.78 | 92.31 | 79.48 | 78.88 | 57.61 | 54.68 |
| | SumGNN | 87.30 | 92.71 | 35.28 | 48.85 | 17.85 | 25.28 | 93.20 | 94.62 | 80.66 | 81.31 | 60.57 | 55.29 |
| | EmerGNN | 94.10 | 97.42 | 62.28 | 68.47 | 27.84 | 46.34 | 96.17 | 96.94 | 89.21 | 90.52 | 81.43 | 79.67 |
| | TIGER | 93.53 | 95.90 | 57.52 | 60.21 | 19.78 | 33.46 | 95.72 | 95.92 | 86.70 | 87.15 | 69.95 | 63.85 |
| Multimodal | Static Fusion | 94.44 | 97.36 | 60.96 | 67.47 | 31.24 | 42.14 | 92.86 | 94.42 | 88.54 | 89.45 | 79.61 | 78.72 |
| | Ensemble | 94.07 | 97.58 | 61.31 | 67.68 | 32.70 | 46.45 | 93.52 | 94.74 | 90.23 | 91.16 | 85.58 | 84.66 |
| | **M²DDI** | **96.26**** | **97.82*** | **68.28*** | **71.73*** | **40.52**** | **46.49** | **98.18**** | **98.86**** | **92.31**** | **93.28*** | **87.64**** | **85.76*** |

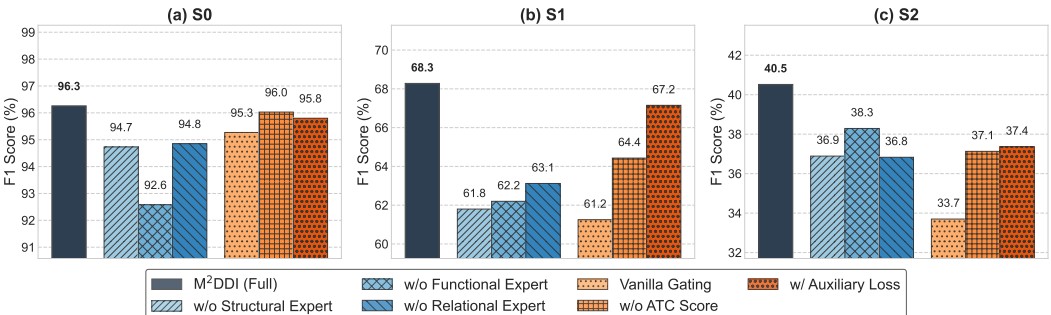

Figure 3: Ablation study on M²DDI, evaluated on the DrugBank dataset from S0 to S2.

that non-adaptive integration is inadequate, emphasizing that the fusion method is as crucial as the modality choice.

M²DDI addresses these challenges through its dynamic, pharmacology-aware gating mechanism, with its advantages most evident in the most demanding inductive settings. For example, on the DrugBank S2 benchmark, M²DDI achieves an F1 score of 40.52, outperforming the next-best Ensemble baseline by nearly 8 points. This enhanced generalization can be attributed to M²DDI's capacity to dynamically prioritize the most reliable evidence source for each drug pair, rather than being constrained by a single-modality perspective or a fixed, suboptimal fusion strategy.

## 4.3 ABLATION STUDY

We conducted a comprehensive ablation study to assess the contribution of each framework component (Figure 3). The removal of any expert type—**Structural**, **Functional**, or **Relational**—resulted in a measurable performance decline, confirming their complementary roles. Omitting structural experts led to the largest drop in F1 score for the inductive S1 setting (–6.48), while excluding relational experts most adversely affected the transductive S0 setting (–5.41), underscoring the distinct importance of each modality.

The pharmacology-aware gating mechanism further proved indispensable for generalization. Substituting it with a vanilla linear gating layer (**Vanilla Gating**) caused severe performance degradation in the S1 (–7.03) and S2 (–6.82) settings, highlighting the superiority of our specialized architecture. Furthermore, the exclusion of the knowledge-based **ATC Score** consistently reduced scores, demonstrating the value of pharmacological priors. Additionally, our loss-free load balancing strat-

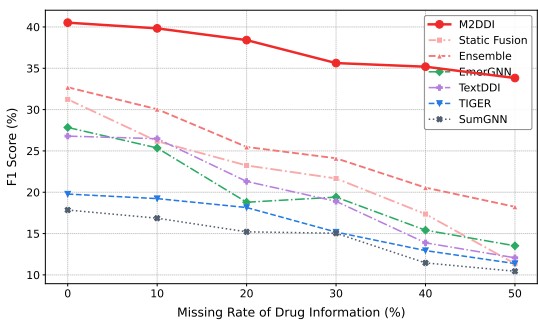 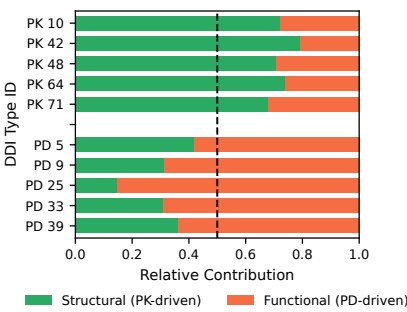

Figure 4: F1 score comparison under increasing data sparsity on DrugBank S2. The comparison includes all fusion-based methods and the top-performing unimodal baseline.

Figure 5: Expert specialization for known PK/PD-driven DDIs on Drug-Bank S0. Full visualization is available in Appendix K.

egy outperformed the traditional **Auxiliary Loss** variant, supporting its effectiveness in managing a heterogeneous expert pool. The Auxiliary Loss refers to the standard mixture-of-experts (MoE) regularization, which encourages uniform expert utilization by penalizing imbalanced expert assignment during training.

Further analyses in the appendix provide detailed validation of our design choices. The ablation of the expert pool configuration supports our architecture by showing that balancing comprehensive evidence with dynamic gating is critical (Appendix H.1). Separately, the examination of the ATC prior underscores the importance of our prior-enhanced gating for achieving strong generalization in clinically relevant, data-scarce scenarios (Appendix H.2).

### 4.4 ROBUSTNESS TO DATA INCOMPLETENESS

The S2 setting challenges models with entirely new drugs, but a more realistic scenario involves new drugs for which we also have scarce information (Gangwal et al., 2024). To simulate this compounded challenge—where drugs are not only novel but their associated functional and relational data are sparse—we designed a targeted robustness evaluation. Within the S2 setting, we progressively masked, for an increasing fraction of drugs, the modalities used by each model—masking only the functional and relational inputs while retaining always-available molecular structures for fusion-based methods, and masking the corresponding single modality for unimodal baselines.

As illustrated in Figure 4, $M^2DDI$ demonstrates substantially greater resilience compared to all baselines. While the performance of all models degrades with increasing data loss, $M^2DDI$ maintains a significant performance advantage. For instance, with functional and relational data missing for $50\%$ of drugs, $M^2DDI$ achieves an F1 score of 33.82, drastically outperforming static fusion (11.35), ensemble (18.24), and even the strongest unimodal baselines. This resilience stems from its dynamic gating mechanism, which learns to intelligently route around missing modalities and rely on the available evidence. Unlike static fusion methods that are crippled by corrupted inputs or unimodal models that fail when their specific data is absent, $M^2DDI$'s dynamic expert selection provides a robust solution to the critical problem of incomplete data in real-world pharmacological databases.

### 4.5 CASE STUDY OF EXPERT SPECIALIZATION

To assess whether our gating mechanism captures pharmacologically meaningful patterns, we analyzed its routing behavior on real-world DrugBank S0 interactions with established mechanisms. We curated clinically recognized DDIs, categorizing each as predominantly pharmacokinetic (PK) or pharmacodynamic (PD) driven based on authoritative literature (Cascorbi, 2012), providing an objective ground truth for model evaluation.

As shown in Figure 5, the model's expert selection quantitatively mirrors pharmacological expectations. For PK-driven interactions—such as those involving metabolic inhibition—the model assigns greater weight to structural experts. In contrast, for PD-driven interactions, including synergistic

or antagonistic target effects, functional experts are prioritized. This correspondence demonstrates that $M^2DDI$ not only integrates multiple modalities, but also learns to identify and prioritize evidence according to underlying pharmacological reasoning, thereby enhancing transparency in its decision-making process.

## 5 CONCLUSION AND FUTURE WORKS

We presented $M^2DDI$, a Mixture-of-Experts framework that reconceptualizes drug-drug interaction prediction as a dynamic selection among specialized experts. By adaptively integrating analyses of pharmacokinetics, pharmacodynamics, and relational pathways, $M^2DDI$ achieves state-of-the-art performance and addresses the generalization challenges faced by prior approaches, particularly in inductive settings. Importantly, the model's gating decisions align with established pharmacological mechanisms, providing interpretable and mechanistically-grounded evidence beyond predictive accuracy. $M^2DDI$ thus advances a robust paradigm for DDI prediction and contributes directly to the development of computational tools for patient safety. Future work could extend this paradigm to the related task of Drug-Target Interaction (DTI) prediction, where the gating mechanism would be adapted to integrate existing drug-focused experts with new experts designed to analyze protein features (e.g., from ESM or AlphaFold) to predict molecular binding and attribute the underlying interaction mechanism.

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

## A   USE OF LARGE LANGUAGE MODELS

We acknowledge the use of a large language model (LLM) as a general-purpose tool for language editing and improving clarity in this manuscript. The scope of its use was strictly confined to stylistic enhancements of the text. All core research contributions—including ideation, methodology, experimentation, and analysis—are the exclusive work of the authors. The authors retain full responsibility for the entire content of this manuscript, and are accountable for its accuracy, originality, and adherence to scientific and ethical standards.

## B   ETHICS STATEMENT

This work is of a technical nature, introducing a new computational framework, $\mathtt{M^2DDI}$, for predicting drug-drug interactions. The research relies exclusively on publicly available, anonymized benchmark datasets (DrugBank and TWOSIDES), and does not involve any experiments with human subjects or the use of private or sensitive patient data. The goal of this research is to advance computational tools for patient safety, and we do not foresee any direct potential for negative societal impact or misuse of this technology. The authors have read and adhered to the ICLR Code of Ethics in the preparation of this manuscript.

## C   PHARMACOLOGICAL MODALITY ENCODING DETAILS

To construct a comprehensive profile for each drug, we process three distinct data modalities, corresponding to different pharmacological perspectives. For the feature-based modalities, structural and functional representations are projected into a unified embedding space of dimension $d_{\text{embed}}$ to enable effective comparison and integration. Concurrently, relational information is captured in the form of a contextual subgraph, designed for graph-based reasoning.

### C.1   STRUCTURAL REPRESENTATION.

A drug's molecular structure is a primary determinant of its pharmacokinetic (PK) properties. For each drug $drug_k$ (where $k \in \{i, j\}$), we employ a *Molecular Graph Encoder*, $f_{\text{struct}}(\cdot)$, to generate a feature vector from its molecular graph. This vector is then mapped into the shared embedding space via a linear projection head $\mathtt{Proj}_{\text{struct}}$.

$$\mathbf{x}_{\text{struct}}^{(k)} = \mathtt{Proj}_{\text{struct}}(f_{\text{struct}}(drug_k)) \in \mathbb{R}^{d_{\text{embed}}}. \tag{10}$$

The structural representation for the drug pair, $\mathbf{x}_{\text{struct}}^{(i,j)}$, is formed by the concatenation of the individual drug embeddings:

$$\mathbf{x}_{\text{struct}}^{(i,j)} = \mathtt{Concat}(\mathbf{x}_{\text{struct}}^{(i)}, \mathbf{x}_{\text{struct}}^{(j)}) \in \mathbb{R}^{2d_{\text{embed}}}. \tag{11}$$

### C.2   FUNCTIONAL REPRESENTATION.

To capture pharmacodynamic (PD) aspects, we leverage a *Textual Information Encoder*, $f_{\text{func}}(\cdot)$, to process rich textual information, $\mathtt{Info}(drug_k)$. A corresponding linear projection head, $\mathtt{Proj}_{\text{func}}$, maps the resulting features into the same shared embedding space.

$$\mathbf{x}_{\text{func}}^{(k)} = \mathtt{Proj}_{\text{func}}(f_{\text{func}}(\mathtt{Info}(drug_k))) \in \mathbb{R}^{d_{\text{embed}}}. \tag{12}$$

Symmetrically, the pair's functional representation is created by concatenation:

$$\mathbf{x}_{\text{func}}^{(i,j)} = \mathtt{Concat}(\mathbf{x}_{\text{func}}^{(i)}, \mathbf{x}_{\text{func}}^{(j)}) \in \mathbb{R}^{2d_{\text{embed}}}. \tag{13}$$

### C.3   RELATIONAL SUBGRAPH FORMULATION.

To situate the drug pair within the broader biomedical landscape, we utilize a large-scale knowledge graph (KG) $\mathcal{G}$. For any given pair $(drug_i, drug_j)$, we extract a localized, path-centric subgraph $\mathcal{G}^{(i,j)}$ that encompasses all directed relational paths from $drug_i$ to $drug_j$ up to a maximum length $L$.

$$\mathcal{G}^{(i,j)} = \mathtt{SubgraphExtractor}(\mathcal{G}, drug_i, drug_j, L). \tag{14}$$

This subgraph $\mathcal{G}^{(i,j)}$ serves as the relational input for the relational expert.

# D  TRAINING OBJECTIVE DETAILS

The entire M$^2$DDI framework is trained end-to-end using a loss function appropriate for the specific dataset's task formulation.

For the **DrugBank** dataset, which is a multi-class classification task, we minimize the standard cross-entropy loss $\mathcal{L}_{\text{CE}}$. For a given drug pair $(drug_i, drug_j)$ with its ground-truth interaction type encoded as a one-hot vector $\mathbf{y}^{(i,j)}$, the loss is calculated as:

$$\mathcal{L} = \mathcal{L}_{\text{CE}}(\mathbf{p}^{(i,j)}, \mathbf{y}^{(i,j)}) = - \sum_{c=1}^{|\mathcal{T}_{\text{DDI}}|} y_c^{(i,j)} \log(p_c^{(i,j)})$$

where $\mathbf{p}^{(i,j)}$ represents the predicted probability distribution obtained by applying the Softmax function to the final logits for all $|\mathcal{T}_{\text{DDI}}|$ interaction types.

For the **TWOSIDES** dataset, which is a multi-label classification task, we treat the prediction for each of the 200 side effects as an independent binary classification problem. The overall loss is the average of the binary cross-entropy (BCE) losses across all possible labels:

$$\mathcal{L} = \frac{1}{|\mathcal{T}_{\text{DDI}}|} \sum_{c=1}^{|\mathcal{T}_{\text{DDI}}|} \mathcal{L}_{\text{BCE}}(p_c^{(i,j)}, y_c^{(i,j)})$$

where $p_c^{(i,j)}$ is the predicted probability for the $c$-th side effect, $y_c^{(i,j)} \in \{0, 1\}$ is the corresponding ground-truth label, and $\mathcal{L}_{\text{BCE}}$ is the BCE loss with logits, defined as:

$$\mathcal{L}_{\text{BCE}}(p, y) = -[y \cdot \log(\sigma(p)) + (1 - y) \cdot \log(1 - \sigma(p))]$$

where $\sigma(\cdot)$ is the sigmoid function. In both cases, all model parameters, including those of the expert networks, the gating mechanism, and the projection heads, are optimized concurrently via this single training objective.

# E  ALGORITHMS FOR RELATIONAL EXPERT

The algorithm for the subgraph extractor and relational expert, presented below.

## E.1  ALGORITHM FOR PATH EXTRACTION.

Given a drug pair $(u, v)$, we use beam search to find the top $B = 5$ paths in both the $u \to v$ and $v \to u$ directions. The procedure for one direction is shown in Algorithm 1. We use three lists: 'openList' for the top entities at each step, 'closeList' for accumulated path scores, and 'pathList' for the paths themselves. First, candidate nodes at each hop distance are identified via bidirectional BFS (lines 3-4). Then, for each step, we compute accumulated path scores using attention weights and use beam search to select the top-$B$ paths to extend in the next step (lines 11-13).

## E.2  ALGORITHM FOR GNN MODULE.

Given the biomedical KG and a drug pair $(u, v)$, it is time-consuming to explicitly extract all paths connecting them. In practice, the pair-wise representations are encoded implicitly using the flow-based GNN detailed in Algorithm 2. Take the direction $u \to v$ as an example. We initialize the representation $\boldsymbol{h}_{u,e}^0 = \boldsymbol{f}_u$ if $e = u$, otherwise $\boldsymbol{h}_{u,e}^0 = \mathbf{0}$. The messages are computed based on a dot product operator $\boldsymbol{h}_{u,e'}^{(\ell-1)} \odot \boldsymbol{h}_r^{(\ell)}$. In this way, the representations of all entities with a path length longer than $\ell$ from $u$ will be $\mathbf{0}$ in the $\ell$-th step. At the end, only entities within a path length of $L$ will have valid representations. This process implicitly encodes the relevant subgraph connecting $u$ and $v$.

---

**Algorithm 1** Path Extractor

---

**Require:** $(u, v), L, B$  {$B$: the number of top paths in each direction.}
1: initialize openList$[0] \leftarrow u$;
2: set $\mathcal{V}_{u,v}^{(0)} = \{u\}, \mathcal{V}_{u,v}^{(L)} = \{v\}$;
3: obtain the set $\mathcal{V}_{u,v}^{(\ell)} = \{e : d(e, u) = \ell, d(e, v) = L - \ell\}, \ell = 1, \ldots, L$ with bread-first-search;
4: **for** $\ell \leftarrow 1$ to $L$ **do**
5:     set closeList$[\ell] \leftarrow \emptyset$, pathList$[\ell] \leftarrow \emptyset$;
6:     **for** each edge in $\{(e', r, e) : e' \in \text{openList}[\ell - 1], e \in \mathcal{V}_{u,v}^{\ell}\}$ **do**
7:         compute the attention weights $\alpha_r^{(\ell)}$;
8:         compute score$(u, e', e) = $ score$(u, e') + \alpha_r^{(\ell)}$;
9:         closeList$[\ell]$.add$((e, \text{score}(u, e', e)))$;
10:     **end for**
11:     **for** $(u, e', e) \in \text{top}_B(\text{closeList}[\ell])$ **do**
12:         openList$[\ell]$.add$(e)$, pathList$[\ell]$.add$((e', r, e))$;
13:     **end for**
14: **end for**
15: **Return:** join(pathList$[1] \ldots$ pathList$[L]$).

---

**Algorithm 2** Relational Expert GNN Module

---

**Require:** $(u, v), L, \delta, \sigma, \{\boldsymbol{W}^{(\ell)}, \boldsymbol{w}^{(\ell)}\}_{\ell=1\ldots L}$.
    $\{(u, v)$: drug pair; $L$: the depth of path-based subgraph; $\delta$: activation function; $\sigma$: sigmoid function; $\{\boldsymbol{W}^{(\ell)}, \boldsymbol{w}^{(\ell)}\}_{\ell=1\ldots L}$: learnable parameters.$\}$
1: initialize the $u \to v$ pair-wise representation as $\boldsymbol{h}_{u,e}^0 = \boldsymbol{f}_u$ if $e = u$, otherwise $\boldsymbol{h}_{u,e}^0 = \boldsymbol{0}$;
2: initialize the $v \to u$ pair-wise representation as $\boldsymbol{h}_{v,e}^0 = \boldsymbol{f}_v$ if $e = v$, otherwise $\boldsymbol{h}_{v,e}^0 = \boldsymbol{0}$;
3: **for** $\ell \leftarrow 1$ to $L$ **do**
4:     **for** $e \in \mathcal{V}_D$ **do** {This loop can work with matrix operations in parallel.}
5:         message for $u \to v$:
        $\boldsymbol{h}_{u,e}^{(\ell)} = \delta \left( \boldsymbol{W}^{(\ell)} \sum_{(e',r,e) \in \mathcal{N}_D} \sigma \left( (\boldsymbol{w}_r^{(\ell)})^\top [\boldsymbol{f}_u; \boldsymbol{f}_v] \right) \cdot \left( \boldsymbol{h}_{u,e'}^{(\ell-1)} \odot \boldsymbol{h}_r^{(\ell)} \right) \right)$;
6:         message for $v \to u$:
        $\boldsymbol{h}_{v,e}^{(\ell)} = \delta \left( \boldsymbol{W}^{(\ell)} \sum_{(e',r,e) \in \mathcal{N}_D} \sigma \left( (\boldsymbol{w}_r^{(\ell)})^\top [\boldsymbol{f}_u; \boldsymbol{f}_v] \right) \cdot \left( \boldsymbol{h}_{v,e'}^{(\ell-1)} \odot \boldsymbol{h}_r^{(\ell)} \right) \right)$;
7:     **end for**
8: **end for**
9: **Return** $\boldsymbol{W}_{\text{rel}}[\boldsymbol{h}_{u,v}^{(L)}; \boldsymbol{h}_{v,u}^{(L)}]$.

---

# F IMPLEMENTATION DETAILS

## F.1 DATASETS AND STATISTICS

We use two benchmark datasets for DDI prediction, following established methodologies from (Zitnik et al., 2018) and (Yu et al., 2021).

- **DrugBank** (Wishart et al., 2018) is a comprehensive drug database. We use the version curated by (Zhang et al., 2023), which defines a multi-class DDI prediction task with 86 distinct interaction types.

- **TWOSIDES** (Tatonetti et al., 2012) is a database of adverse drug reactions derived from clinical reports. We use a pre-processed version that frames the task as a multi-label prediction problem over 200 common side effects.

To evaluate model generalization, we split the data according to three settings (S0, S1, S2). For the S1 and S2 settings, which involve emerging drugs, the set of all drugs $\mathcal{V}_D$ is randomly split into three disjoint sets: $\mathcal{V}_D = \mathcal{V}_{\text{D-train}} \cup \mathcal{V}_{\text{D-valid}} \cup \mathcal{V}_{\text{D-test}}$, where drugs in $\mathcal{V}_{\text{D-valid}}$ and $\mathcal{V}_{\text{D-test}}$ are considered emerging.

| Hyperparameter | Setting | DrugBank | TWOSIDES |
|---|---|---|---|
| ***General Model Architecture*** | | | |
| Embedding dimension ($d_{\mathrm{embed}}$) | | 256 | |
| Hidden dimension ($d_{\mathrm{hidden}}$) | | 128 | |
| ATC embedding dimension ($d_{\mathrm{ATC}}$) | | 128 | |
| No. of Structural Experts ($N_s$) | All | 4 | |
| No. of Functional Experts ($N_f$) | | 4 | |
| No. of Relational Experts ($N_r$) | | 1 | |
| No. of activated experts ($K$) | | 5 | |
| ***Relational Expert Details*** | | | |
| Max KG path length ($L$) | All | 3 | |
| Path extractor beam width ($B$) | | 5 | |
| ***Training Details*** | | | |
| Optimizer | | Adam | |
| Dropout rate | All | 0.3 | |
| Epochs | | 100 | |
| Load balancing step size ($\alpha$) | | 1e-3 | |
| | S0 | 0.001 | 0.01 |
| Learning rate | S1 | 0.001 | 0.001 |
| | S2 | 0.001 | 0.003 |
| | S0 | 128 | 32 |
| Batch size | S1 | 32 | 32 |
| | S2 | 32 | 64 |

Table 3: Detailed hyperparameters for the $\mathrm{M^2DDI}$ framework, specified by dataset and experimental setting. General architectural parameters are consistent across all settings, while training parameters are tuned for each specific task.

- In the **S0** setting, all drugs are known, and interactions are split into training, validation, and testing sets.
- In the **S1** setting, test interactions occur between a known drug (from $\mathcal{V}_{\mathrm{D\text{-}train}}$) and an emerging drug (from $\mathcal{V}_{\mathrm{D\text{-}test}}$).
- In the **S2** setting, test interactions occur between two emerging drugs (both from $\mathcal{V}_{\mathrm{D\text{-}test}}$).

Detailed statistics for each data split are provided in Table 4 and Table 5. For the TWOSIDES dataset, we follow (Yu et al., 2021) to create negative samples during evaluation by randomly pairing an emerging drug with a known drug, ensuring the resulting interaction does not exist in the ground-truth set.

| Dataset | $|\mathcal{V}_{\mathrm{D}}|$ | $|\mathcal{R}_{\mathrm{I}}|$ | $|\mathcal{N}_{\mathrm{D\text{-}train}}|$ | $|\mathcal{N}_{\mathrm{D\text{-}valid}}|$ | $|\mathcal{N}_{\mathrm{D\text{-}test}}|$ |
|---|---|---|---|---|---|
| DrugBank | 1,710 | 86 | 134,641 | 19,224 | 38,419 |
| TWOSIDES | 604 | 200 | 177,568 | 24,887 | 49,656 |

Table 4: Dataset statistics for the S0 (transductive) setting.

## F.2 EVALUATION METRICS

We evaluate model performance using standard metrics appropriate for each task, following established practices (Yu et al., 2021; Zitnik et al., 2018).

**For DrugBank (Multi-class Classification)**  As noted by (Yu et al., 2021), the DrugBank dataset contains at most one interaction type per drug pair. We therefore evaluate performance in a multi-class setting. The metrics are:

- **Accuracy**: The proportion of correctly predicted interaction types compared to the ground-truth types.

| Dataset | Seed | $|\mathcal{V}_{\text{D-train}}|$ | $|\mathcal{V}_{\text{D-valid}}|$ | $|\mathcal{V}_{\text{D-test}}|$ | $|\mathcal{R}_{\text{I}}|$ | $|\mathcal{N}_{\text{D-train}}|$ | S1 Setting | | S2 Setting | |
|---|---|---|---|---|---|---|---|---|---|---|
| | | | | | | | $|\mathcal{N}_{\text{D-valid}}|$ | $|\mathcal{N}_{\text{D-test}}|$ | $|\mathcal{N}_{\text{D-valid}}|$ | $|\mathcal{N}_{\text{D-test}}|$ |
| DrugBank | 1 | 1,461 | 79 | 161 | 86 | 137,864 | 17,591 | 32,322 | 536 | 1,901 |
| | 12 | 1,465 | 79 | 161 | 86 | 140,085 | 17,403 | 30,731 | 522 | 1,609 |
| | 123 | 1,466 | 81 | 161 | 86 | 140,353 | 14,933 | 32,845 | 396 | 1,964 |
| | 1234 | 1,463 | 81 | 162 | 86 | 139,141 | 15,635 | 33,254 | 434 | 1,956 |
| | 12345 | 1,461 | 80 | 169 | 86 | 133,394 | 17,784 | 35,803 | 546 | 2,355 |
| TWOSIDES | 1 | 514 | 30 | 60 | 200 | 185,673 | 16,113 | 45,365 | 467 | 2,466 |
| | 12 | 514 | 30 | 60 | 200 | 172,351 | 23,815 | 48,638 | 717 | 3,373 |
| | 123 | 514 | 30 | 60 | 200 | 181,257 | 18,209 | 46,969 | 358 | 2,977 |
| | 1234 | 514 | 30 | 60 | 200 | 186,104 | 25,830 | 35,302 | 837 | 1,605 |
| | 12345 | 514 | 30 | 60 | 200 | 179,993 | 22,059 | 43,867 | 702 | 2,695 |

Table 5: Dataset statistics for S1 and S2 (inductive) settings across 5 different random seeds.

- **Macro F1-Score**: The unweighted arithmetic mean of the F1-scores for each interaction type. This metric is crucial for imbalanced datasets as it treats all classes equally. It is defined as:

$$\text{F1}_{\text{macro}} = \frac{1}{|\mathcal{R}_{\text{I}}|} \sum_{i \in \mathcal{R}_{\text{I}}} \frac{2 \times P_i \times R_i}{P_i + R_i}$$

  where $P_i$ and $R_i$ are the precision and recall for interaction type $i$, respectively, and $|\mathcal{R}_{\text{I}}|$ is the total number of interaction types.

- **Cohen's Kappa** (Cohen, 1960): $\kappa = \frac{A_p - A_e}{1 - A_e}$, where $A_p$ is the observed agreement (accuracy) and $A_e$ is the probability of randomly seeing each class.

**For TWOSIDES (Multi-label Classification)** In the TWOSIDES dataset, a drug pair can be associated with multiple side effects (e.g., anaemia, nausea). We thus model this as a multi-label task, where each side effect is an independent binary classification problem. For evaluation, we sample one negative drug pair for each positive test instance and report the following metrics, averaged across all side effect types:

- **ROC-AUC**: The Area Under the Receiver Operating Characteristic Curve. It measures the trade-off between the true positive rate and false positive rate across different thresholds. It is calculated as $\sum_{k=1}^{n} \text{TP}_k \Delta \text{FP}_k$, where $(\text{TP}_k, \text{FP}_k)$ is the true positive rate and false positive rate at the $k$-th operating point.

- **PR-AUC**: The Area Under the Precision-Recall Curve. This metric is particularly informative for imbalanced tasks, as it evaluates the trade-off between precision and recall. It is measured by $\sum_{k=1}^{n} P_k \Delta R_k$, where $(P_k, R_k)$ is the precision-recall pair at the $k$-th operating point.

- **Accuracy**: the average precision of drug pairs for each side effect.

### F.3 BASELINE METHODS

We compare M²DDI against a suite of baselines that represent the primary paradigms in DDI prediction.

- **Structure-Centric**:
  - **MLP** (Rogers & Hahn, 2010): A standard multilayer perceptron that uses pre-computed molecular fingerprints as input features.
  - **DeepDDI** (Ryu et al., 2018): A deep learning model that processes drug SMILES strings to learn structural feature representations for prediction.
- **Function-Centric**:
  - **TextDDI** (Zhu et al., 2023): A model that uses a pre-trained language model to encode textual descriptions of drugs (e.g., indications, mechanism-of-action).
- **Relation-Centric**:
  - **Decagon** (Zitnik et al., 2018): A pioneering graph neural network model for DDI prediction on a large, multi-relational graph.

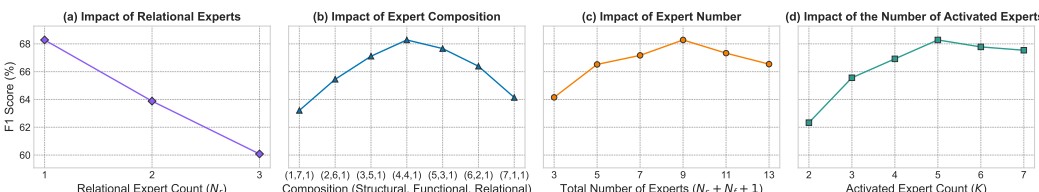

Figure 6: Analysis of expert pool configuration (DrugBank S1, $N_r = 1$ fixed): (a) Structural/functional expert composition ($N_s + N_f = 8$, $K = 5$). (b) Total expert pool size ($N_s + N_f + 1$, balanced composition, $K = 5$). (c) Activated expert count ($K$) for optimal 9-expert pool (4+4+1).

- **SumGNN** (Yu et al., 2021): A GNN-based model that samples a subgraph for each drug pair and uses a summarization mechanism to extract reasoning paths.
- **EmerGNN** (Zhang et al., 2023): A model that designs a flow-based GNN to learn subgraph representations for DDI prediction.
- **TIGER** (Su et al., 2024): A graph transformer model that learns dual-channel representations from both molecular structures and a biomedical KG.

### F.4 HYPERPARAMETERS AND COMPUTING INFRASTRUCTURE

All experiments were conducted on a server equipped with one NVIDIA A6000 GPU. The software environment consists of CUDA 11.3.1, Python 3.8, and PyTorch 1.10.1. Key hyperparameters for the `M`$^2$`DDI` model, which vary based on the dataset and experimental setting, are detailed in Table 3. With this setup, the training for all settings could be completed within 24 hours.

## G FULL EXPERIMENTAL RESULTS

This section provides the comprehensive numerical results of our experiments, extending the analysis presented in the main paper. We evaluate the performance of `M`$^2$`DDI` against various baseline methods across three distinct experimental settings (S0, S1, and S2) on both the DrugBank and TWOSIDES datasets.

Table 6 details the performance using standard evaluation metrics: F1-score and Accuracy for DrugBank, and PR-AUC and ROC-AUC for TWOSIDES. The baselines are categorized into Structure-based (MLP, DeepDDI), Function-based (TextDDI), Relation-based (Decagon, SumGNN, EmerGNN, TIGER), and Multimodal methods (Static Fusion, Ensemble). As observed, `M`$^2$`DDI` consistently achieves the best performance across all metrics and data splits. Notably, in the more challenging Inductive settings (S1 and S2), our method demonstrates a significant margin of improvement over the second-best performing baselines (e.g., EmerGNN and Ensemble). The statistical significance of these improvements is verified by paired t-tests, with $p$-values reported in the final row, most of which are well below $0.05$.

To further validate the robustness of our model, we report additional evaluation metrics in Table 7. Specifically, we utilize Cohen's Kappa for the DrugBank dataset to account for class imbalance, and standard Accuracy for the TWOSIDES dataset. Consistent with the primary results, `M`$^2$`DDI` outperforms all competing methods in these supplementary metrics. For instance, in the S0 split of DrugBank, our model achieves a Cohen's Kappa of 97.45, surpassing the strongest multimodal baseline. These results reinforce the conclusion that `M`$^2$`DDI` effectively integrates multi-view information to predict drug-drug interactions accurately, even in complex, unseen scenarios.

## H MORE ABLATION STUDIES

### H.1 ANALYSIS OF EXPERT POOL CONFIGURATION

We systematically evaluate how the configuration of the expert pool influences `M`$^2$`DDI` performance in the DrugBank S1 setting, validating our final architectural decisions through four analyses. First, we find that employing a single relational expert ($N_r = 1$) is optimal (Figure 6a), as additional

| Category | Method | DrugBank | | | | | | TWOSIDES | | | | | |
|---|---|---|---|---|---|---|---|---|---|---|---|---|---|
| | | S0 | | S1 | | S2 | | S0 | | S1 | | S2 | |
| | | F1 | Acc | F1 | Acc | F1 | Acc | PR-AUC | ROC-AUC | PR-AUC | ROC-AUC | PR-AUC | ROC-AUC |
| Structure | MLP | 60.75 (± 0.44) | 82.57 (± 0.30) | 21.52 (± 0.83) | 46.54 (± 2.30) | 20.15 (± 2.98) | 39.85 (± 5.25) | 81.71 (± 0.11) | 81.82 (± 0.27) | 81.70 (± 1.54) | 81.61 (± 2.29) | 64.84 (± 5.17) | 58.52 (± 6.47) |
| | DeepDDI | 73.55 (± 0.25) | 88.26 (± 0.09) | 34.36 (± 2.05) | 36.31 (± 2.67) | 13.73 (± 2.90) | 15.86 (± 5.17) | 89.69 (± 0.04) | 91.85 (± 0.05) | 83.06 (± 1.56) | 83.76 (± 2.35) | 68.51 (± 5.70) | 62.13 (± 6.15) |
| Function | TextDDI | 92.00 (± 0.56) | 95.75 (± 0.49) | 59.51 (± 1.08) | 66.40 (± 0.50) | 26.79 (± 1.00) | 44.23 (± 0.57) | 92.67 (± 0.61) | 94.56 (± 0.49) | 84.29 (± 0.57) | 85.83 (± 1.10) | 83.18 (± 0.58) | 78.16 (± 1.08) |
| Relation | Decagon | 56.91 (± 0.40) | 86.97 (± 0.23) | 24.66 (± 4.26) | 47.63 (± 4.97) | 6.12 (± 2.27) | 22.47 (± 3.30) | 90.78 (± 0.36) | 92.31 (± 0.37) | 79.48 (± 2.25) | 78.88 (± 2.77) | 57.61 (± 5.77) | 54.68 (± 6.26) |
| | SumGNN | 87.30 (± 0.39) | 92.71 (± 0.08) | 35.28 (± 3.86) | 48.85 (± 8.92) | 17.85 (± 2.15) | 25.28 (± 2.96) | 93.20 (± 0.16) | 94.62 (± 0.46) | 80.66 (± 1.30) | 81.31 (± 1.12) | 60.57 (± 8.60) | 55.29 (± 6.69) |
| | EmerGNN | 94.10 (± 0.74) | 97.42 (± 0.06) | 62.28 (± 2.10) | 68.47 (± 1.47) | 27.84 (± 3.06) | 46.34 (± 3.46) | 96.17 (± 0.10) | 96.94 (± 0.09) | 89.21 (± 0.74) | 90.52 (± 1.46) | 81.43 (± 7.52) | 79.67 (± 9.51) |
| | TIGER | 93.53 (± 0.35) | 95.90 (± 0.36) | 57.52 (± 1.96) | 60.21 (± 2.73) | 19.78 (± 2.05) | 33.46 (± 3.58) | 95.72 (± 0.23) | 95.92 (± 0.10) | 86.70 (± 0.60) | 87.15 (± 3.81) | 69.95 (± 7.00) | 63.85 (± 5.68) |
| Multimodal | Static Fusion | 94.44 (± 0.30) | 97.36 (± 0.35) | 60.96 (± 4.20) | 67.47 (± 2.75) | 31.24 (± 1.37) | 42.14 (± 5.28) | 92.86 (± 0.21) | 94.42 (± 0.62) | 88.54 (± 1.09) | 89.45 (± 3.00) | 79.61 (± 2.83) | 78.72 (± 0.92) |
| | Ensemble | 94.07 (± 0.34) | 97.58 (± 0.11) | 61.31 (± 4.83) | 67.68 (± 2.85) | 32.70 (± 1.75) | 46.45 (± 0.23) | 93.52 (± 0.47) | 94.74 (± 0.19) | 90.23 (± 0.91) | 91.16 (± 0.24) | 85.58 (± 0.91) | 84.66 (± 0.92) |
| | $M^2$DDI | **96.26** (± **0.36**) | **97.82** (± **0.13**) | **68.28** (± **3.36**) | **71.73** (± **1.90**) | **40.52** (± **1.28**) | **46.49** (± **0.12**) | **98.18** (± **0.22**) | **98.86** (± **0.11**) | **92.31** (± **0.37**) | **93.28** (± **1.16**) | **87.64** (± **0.19**) | **85.76** (± **0.52**) |
| p-value | | $2.93 \times 10^{-5}$ | $1.40 \times 10^{-2}$ | $1.24 \times 10^{-2}$ | $1.74 \times 10^{-2}$ | $6.74 \times 10^{-5}$ | $7.42 \times 10^{-1}$ | $3.08 \times 10^{-6}$ | $2.82 \times 10^{-9}$ | $4.48 \times 10^{-3}$ | $1.37 \times 10^{-2}$ | $6.22 \times 10^{-3}$ | $5.67 \times 10^{-2}$ |

Table 6: Comprehensive performance comparison across all settings on the DrugBank and TWO-SIDES datasets. For each metric, the mean and standard deviation (from 5-fold cross-validation) are reported in the format of mean ± std. The best performance is boldfaced and the second-best is underlined. The final row indicates the p-value of a paired t-test between the best and second-best performing methods.

| Category | Method | DrugBank (Cohen's Kappa) | | | TWOSIDES (Accuracy) | | |
|---|---|---|---|---|---|---|---|
| | | S0 | S1 | S2 | S0 | S1 | S2 |
| Structure | MLP | 80.17 (± 1.00) | 33.26 (± 2.60) | 23.49 (± 4.69) | 73.58 (± 0.23) | 76.39 (± 2.07) | 59.81 (± 5.59) |
| | DeepDDI | 80.91 (± 0.92) | 35.82 (± 3.71) | 14.51 (± 2.95) | 85.18 (± 0.21) | 72.46 (± 2.26) | 55.62 (± 4.61) |
| Function | TextDDI | 95.22 (± 0.47) | 63.52 (± 1.01) | 30.64 (± 1.08) | 87.82 (± 0.58) | 81.89 (± 0.49) | 76.34 (± 1.00) |
| Relation | Decagon | 85.74 (± 1.20) | 35.96 (± 6.85) | 11.43 (± 2.39) | 82.50 (± 0.64) | 69.71 (± 6.15) | 54.07 (± 8.86) |
| | SumGNN | 90.28 (± 1.17) | 40.86 (± 5.34) | 17.25 (± 2.95) | 89.01 (± 0.22) | 73.31 (± 4.49) | 54.36 (± 5.69) |
| | EmerGNN | 96.48 (± 0.97) | 61.91 (± 4.94) | 31.95 (± 3.55) | 92.38 (± 0.22) | 83.03 (± 4.74) | 73.13 (± 7.64) |
| | TIGER | 95.13 (± 1.02) | 58.93 (± 4.68) | 24.34 (± 2.12) | 91.24 (± 0.27) | 79.76 (± 3.00) | 60.74 (± 7.63) |
| Multimodal | Static Fusion | 96.85 (± 1.19) | 65.47 (± 5.10) | 28.53 (± 3.35) | 87.85 (± 0.34) | 80.93 (± 1.47) | 72.01 (± 3.59) |
| | Ensemble | 96.38 (± 0.91) | 65.24 (± 3.55) | 31.65 (± 3.99) | 89.23 (± 0.35) | 82.49 (± 2.33) | 71.94 (± 3.35) |
| | $M^2$DDI | **97.45** (± **1.17**) | **65.67** (± **4.35**) | **33.95** (± **2.10**) | **94.20** (± **0.22**) | **85.53** (± **1.56**) | **77.53** (± **2.26**) |

Table 7: Performance comparison on additional metrics. We report Cohen's Kappa for DrugBank and Accuracy for TWOSIDES across different splits (S0, S1, S2).

GNNs introduce redundant pathway information. Second, with $N_r = 1$ fixed, a balanced (4, 4) composition of structural and functional experts is essential for representing both pharmacokinetic (PK) and pharmacodynamic (PD) perspectives equally (Figure 6b). Third, expanding this balanced

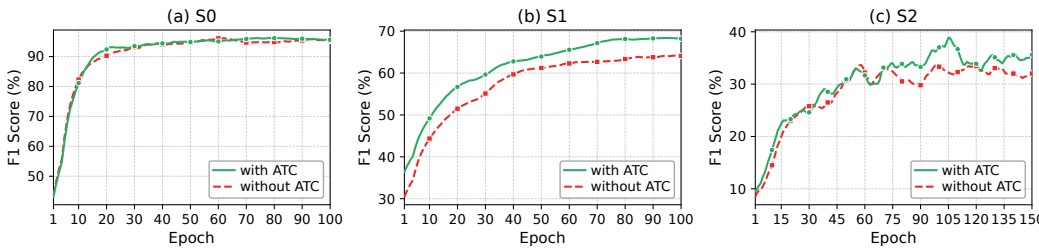

Figure 7: Ablation study of the ATC prior in $\text{M}^2\text{DDI}$, evaluated on the DrugBank dataset. The plots compare the validation F1-score progression of the full model against a variant without the ATC prior path across scenarios S0, S1, and S2.

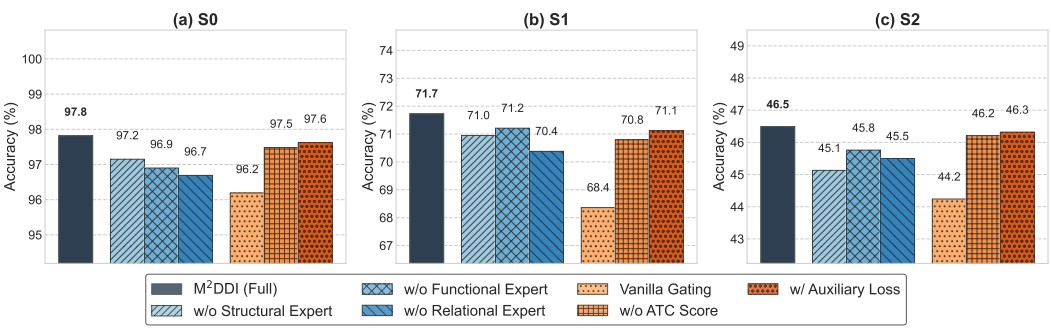

Figure 8: Ablation study on $\text{M}^2\text{DDI}$ evaluated on the DrugBank dataset (S0 to S2) using the accuracy metric.

pool to nine experts ($N_s = 4, N_f = 4, N_r = 1$) achieves peak performance, underscoring the importance of expert diversity (Figure 6c). Finally, for the optimal 9-expert configuration, Figure 6d shows that activating the top $K = 5$ experts—approximately half the pool—yields the best results. These findings indicate that balancing comprehensive evidence integration with dynamic gating is critical, thereby supporting our architectural choices.

### H.2 ABLATION STUDY ON THE ATC PRIOR

To validate the contribution of the biomedical knowledge priors integrated into our gating mechanism, we performed an ablation study by removing the ATC prior path. We compared the performance of the full $\text{M}^2\text{DDI}$ framework against this ablated variant ("without ATC") across the S0, S1, and S2 data-splitting scenarios.

The results, shown in Figure 7, highlight the critical role of this prior knowledge in challenging generalization settings. In the S0 scenario, where all drug features are available during training, the performance difference between the two models is negligible. This indicates that the feature-query path is sufficient for effective expert routing when data is abundant.

Conversely, a pronounced performance gap emerges in the S1 and S2 scenarios, which simulate the prediction of DDIs involving novel drugs. In these cases, the full $\text{M}^2\text{DDI}$ model consistently outperforms the ablated version. This demonstrates that the ATC prior provides a robust and essential signal for the gating mechanism, effectively guiding expert selection when modality-specific features are sparse or entirely absent for new compounds. These findings underscore the importance of our prior-enhanced gating design for achieving strong generalization and reliable performance in clinically relevant, data-scarce scenarios.

### H.3 ABLATION RESULTS IN ACCURACY

Here, we provide additional ablation results in Section 4.3 using the accuracy metric, as shown in Figure 8.

## I  COMPUTATIONAL COMPLEXITY ANALYSIS

The design of the $\texttt{M}^2\texttt{DDI}$ framework prioritizes computational efficiency by leveraging the parallelizable nature of the Mixture-of-Experts (MoE) architecture. The overall complexity is determined by its two main components: the gating mechanism and the heterogeneous expert pool. Our analysis demonstrates that the framework's inference latency is asymptotically bounded by that of its most computationally intensive component, the relational expert.

**Gating Mechanism Complexity.**  The gating mechanism's computation is lightweight. It involves three steps: (1) encoding features, whose complexity depends on the pre-trained encoders but is performed only once per input; (2) calculating feature-based scores ($\mathbf{s}_{\text{feat}}$) and ATC-based scores ($\mathbf{s}_{\text{ATC}}$) via matrix multiplications, with complexities of $O(d_{\text{query}} \cdot N)$ and $O(d_{\text{ATC}} \cdot N)$ respectively; and (3) synthesizing these scores with a final routing matrix, which has a complexity of $O(N^2)$. Since the number of experts $N$ and the embedding dimensions ($d_{\text{query}}, d_{\text{ATC}}$) are small constants, the computational overhead of the gating mechanism is negligible compared to the expert computations.

**Expert Pool Complexity.**  A key advantage of the MoE architecture is that all $N$ experts in the pool can be evaluated in parallel. Therefore, the total wall-clock time (latency) for the expert pool is determined not by the sum of all expert computations, but by the single most time-consuming expert activated for a given input. We analyze the complexity of each expert type:

- **Structural and Functional Experts:** These experts are two-layer MLPs. The complexity for processing a concatenated drug pair representation of size $2d_{\text{embed}}$ is $O(d_{\text{embed}} \cdot d_{\text{hidden}})$. This computation is highly efficient.
- **Relational Expert:** This GNN-based expert is the most computationally intensive component. As detailed in Algorithm 2, its complexity is dictated by the message passing over the knowledge graph. For a path length of $L$, the complexity is approximately $O(L \cdot |\mathcal{E}| \cdot d_{\text{hidden}}^2)$, where $|\mathcal{E}|$ is the number of edges in the biomedical KG.

The computational cost is overwhelmingly dominated by the relational expert. In a typical biomedical KG like HetioNet, the number of edges $|\mathcal{E}|$ is in the millions, whereas embedding dimensions like $d_{\text{embed}}$ and $d_{\text{hidden}}$ are typically a few hundred. Consequently, the GNN's complexity scales with the graph size, making it several orders of magnitude greater than that of the MLP experts: $O(L \cdot |\mathcal{E}| \cdot d_{\text{hidden}}^2) \gg O(d_{\text{embed}} \cdot d_{\text{hidden}})$.

**Overall Complexity and Inference Time.**  During inference, the total latency is the sum of the gating latency and the latency of the slowest expert in the parallel execution.

$$\text{Latency}_{\text{inference}} = \text{Latency}_{\text{gating}} + \max_{i \in \mathcal{I}}(\text{Latency}_{E_i}).$$

Given the analysis above, the 'max' term is always the latency of the relational expert, $E_{\text{rel}}$, whenever it is selected. Therefore, the worst-case inference latency is asymptotically bounded by the relational expert's computation:

$$\text{Latency}_{\text{inference}}^{\text{worst-case}} \approx \text{Latency}_{E_{\text{rel}}}.$$

The relational expert in $\texttt{M}^2\texttt{DDI}$ adopts the flow-based GNN architecture from EmerGNN (Zhang et al., 2023). As a result, its computational complexity is asymptotically identical to that of the EmerGNN model. This analysis conclusively shows that $\texttt{M}^2\texttt{DDI}$ integrates multiple pharmacological perspectives to achieve superior performance and interpretability **without incurring an asymptotic increase in computational cost** over strong, single-modality relational baselines. The framework effectively reaps the benefits of an ensemble while maintaining the computational profile of its most complex member.

## J  EMPIRICAL EFFICIENCY ANALYSIS

To empirically validate the theoretical efficiency discussed in Section I, we conducted a comparative analysis of training time, inference latency, and model parameters. The results, presented in Table 8, demonstrate that M$^2$DDI achieves a highly competitive performance profile.

Our key observations from the experiments are as follows:

Table 8: Efficiency comparison on *DrugBank* and *TWOSIDES*. Lower is better for time and parameter count.

| Method | DrugBank | | | TWOSIDES | | |
|---|---|---|---|---|---|---|
| | Train/epoch (min) | Inference (ms) | Params (M) | Train/epoch (min) | Inference (ms) | Params (M) |
| EmerGNN | 28.0 | 58.64 | 0.14 | 4.1 | 67.54 | 0.16 |
| TextDDI | 107.7 | 38.17 | 124.71 | 94.2 | 41.79 | 355.56 |
| Tiger | 12.3 | 11.05 | 2.52 | 2.0 | 13.41 | 2.54 |
| M$^2$DDI (Ours) | 11.8 | 59.35 | 1.98 | 1.5 | 68.25 | 1.92 |

- **Training Time:** M$^2$DDI demonstrates exceptional training efficiency, achieving the fastest training time per epoch on both datasets. It is significantly faster than the relational baseline (EmerGNN) and the text-based model (TextDDI), and comparable to the highly optimized Tiger model. This efficiency stems from the MoE architecture, which avoids processing all modalities for every sample.

- **Inference Time:** The inference latency of M$^2$DDI is nearly identical to that of EmerGNN, the standalone relational expert model. This result provides strong empirical evidence for our theoretical analysis in Section I, which posits that the overall inference time is bounded by the most computationally intensive expert (i.e., the relational expert). M$^2$DDI successfully incorporates multimodal benefits without incurring a significant penalty in inference speed compared to its most complex component.

- **Parameter Count:** While more complex than the single-modality EmerGNN, M$^2$DDI maintains a modest parameter count, especially when compared to large text-based models like TextDDI. This highlights the parameter-efficient nature of our framework.

In summary, the empirical results confirm that M$^2$DDI is not only effective but also computationally efficient. It offers a superior trade-off between predictive performance and computational cost, making it a practical and scalable solution for DDI prediction.

## K COMPREHENSIVE EXPERT ACTIVATION VISUALIZATION

This section provides a comprehensive visualization of the expert activation profiles, extending Figure 4 from the main paper. Table 9 enumerates all 86 distinct drug-drug interaction types from the DrugBank dataset and their corresponding numerical IDs. Figure 9 then illustrates the relative contributions of structural (PK-focused) and functional (PD-focused) experts for every DDI type in the dataset. The consistent patterns observed—where the gating mechanism's allocations align with the likely underlying pharmacology of an interaction (e.g., prioritizing structural experts for metabolic interactions)—validate the model's interpretability and its ability to perform mechanistic reasoning across the full range of DDI classes.

| ID | DDI Description | ID | DDI Description |
|---|---|---|---|
| 0 | Drug A may increase the photosensitizing activities of Drug B. | 43 | Drug A may increase the central neurotoxic activities of Drug B. |
| 1 | Drug A may increase the anticholinergic activities of Drug B. | 44 | Drug A may decrease effectiveness of Drug B as a diagnostic agent. |
| 2 | The bioavailability of Drug B can be decreased when combined with Drug A. | 45 | Drug A may increase the bronchoconstrictory activities of Drug B. |
| 3 | The metabolism of Drug B can be increased when combined with Drug A. | 46 | The metabolism of Drug B can be decreased when combined with Drug A. |
| 4 | Drug A may decrease the vasoconstricting activities of Drug B. | 47 | Drug A may increase the myopathic rhabdomyolysis activities of Drug B. |
| 5 | Drug A may increase the anticoagulant activities of Drug B. | 48 | The risk or severity of adverse effects can be increased when Drug A is combined with Drug B. |

Table 9 – continued from previous page

| ID | DDI Description | ID | DDI Description |
|---|---|---|---|
| 6 | Drug A may increase the ototoxic activities of Drug B. | 49 | The risk or severity of heart failure can be increased when Drug B is combined with Drug A. |
| 7 | The therapeutic efficacy of Drug B can be increased when used in combination with Drug A. | 50 | Drug A may increase the hypercalcemic activities of Drug B. |
| 8 | Drug A may increase the hypoglycemic activities of Drug B. | 51 | Drug A may decrease the analgesic activities of Drug B. |
| 9 | Drug A may increase the antihypertensive activities of Drug B. | 52 | Drug A may increase the antiplatelet activities of Drug B. |
| 10 | The serum concentration of the active metabolites of Drug B can be reduced when Drug B is used in combination with Drug A resulting in a loss in efficacy. | 53 | Drug A may increase the bradycardic activities of Drug B. |
| 11 | Drug A may decrease the anticoagulant activities of Drug B. | 54 | Drug A may increase the hyponatremic activities of Drug B. |
| 12 | The absorption of Drug B can be decreased when combined with Drug A. | 55 | The risk or severity of hypotension can be increased when Drug A is combined with Drug B. |
| 13 | Drug A may decrease the bronchodilatory activities of Drug B. | 56 | Drug A may increase the nephrotoxic activities of Drug B. |
| 14 | Drug A may increase the cardiotoxic activities of Drug B. | 57 | Drug A may decrease the cardiotoxic activities of Drug B. |
| 15 | Drug A may increase the central nervous system depressant (CNS depressant) activities of Drug B. | 58 | Drug A may increase the ulcerogenic activities of Drug B. |
| 16 | Drug A may decrease the neuromuscular blocking activities of Drug B. | 59 | Drug A may increase the hypotensive activities of Drug B. |
| 17 | Drug A can cause an increase in the absorption of Drug B resulting in an increased serum concentration and potentially a worsening of adverse effects. | 60 | Drug A may decrease the stimulatory activities of Drug B. |
| 18 | Drug A may increase the vasoconstricting activities of Drug B. | 61 | The bioavailability of Drug B can be increased when combined with Drug A. |
| 19 | Drug A may increase the QTc-prolonging activities of Drug B. | 62 | Drug A may increase the myelosuppressive activities of Drug B. |
| 20 | Drug A may increase the neuromuscular blocking activities of Drug B. | 63 | Drug A may increase the serotonergic activities of Drug B. |
| 21 | Drug A may increase the adverse neuromuscular activities of Drug B. | 64 | Drug A may increase the excretion rate of Drug B which could result in a lower serum level and potentially a reduction in efficacy. |
| 22 | Drug A may increase the stimulatory activities of Drug B. | 65 | The risk or severity of bleeding can be increased when Drug A is combined with Drug B. |
| 23 | Drug A may increase the hypocalcemic activities of Drug B. | 66 | Drug A can cause a decrease in the absorption of Drug B resulting in a reduced serum concentration and potentially a decrease in efficacy. |
| 24 | Drug A may increase the atrioventricular blocking (AV block) activities of Drug B. | 67 | Drug A may increase the hyperkalemic activities of Drug B. |
| 25 | Drug A may decrease the antiplatelet activities of Drug B. | 68 | Drug A may increase the analgesic activities of Drug B. |
| 26 | Drug A may increase the neuroexcitatory activities of Drug B. | 69 | The therapeutic efficacy of Drug B can be decreased when used in combination with Drug A. |

Table 9 – continued from previous page

| ID | DDI Description | ID | DDI Description |
|----|----------------|----|----------------|
| 27 | Drug A may increase the dermatologic adverse activities of Drug B. | 70 | Drug A may increase the hypertensive activities of Drug B. |
| 28 | Drug A may decrease the diuretic activities of Drug B. | 71 | Drug A may decrease the excretion rate of Drug B which could result in a higher serum level. |
| 29 | Drug A may increase the orthostatic hypotensive activities of Drug B. | 72 | The serum concentration of Drug B can be increased when it is combined with Drug A. |
| 30 | The risk or severity of hypertension can be increased when Drug B is combined with Drug A. | 73 | Drug A may increase the fluid retaining activities of Drug B. |
| 31 | Drug A may increase the sedative activities of Drug B. | 74 | The serum concentration of Drug B can be decreased when it is combined with Drug A. |
| 32 | The risk or severity of QTc prolongation can be increased when Drug A is combined with Drug B. | 75 | Drug A may decrease the sedative activities of Drug B. |
| 33 | Drug A may increase the immunosuppressive activities of Drug B. | 76 | The serum concentration of the active metabolites of Drug B can be increased when Drug B is used in combination with Drug A. |
| 34 | Drug A may increase the neurotoxic activities of Drug B. | 77 | Drug A may increase the hyperglycemic activities of Drug B. |
| 35 | Drug A may increase the antipsychotic activities of Drug B. | 78 | Drug A may increase the central nervous system depressant (CNS depressant) and hypertensive activities of Drug B. |
| 36 | Drug A may decrease the antihypertensive activities of Drug B. | 79 | Drug A may increase the hepatotoxic activities of Drug B. |
| 37 | Drug A may increase the vasodilatory activities of Drug B. | 80 | Drug A may increase the thrombogenic activities of Drug B. |
| 38 | Drug A may increase the constipating activities of Drug B. | 81 | Drug A may increase the arrhythmogenic activities of Drug B. |
| 39 | Drug A may increase the respiratory depressant activities of Drug B. | 82 | Drug A may increase the hypokalemic activities of Drug B. |
| 40 | Drug A may increase the hypotensive and central nervous system depressant (CNS depressant) activities of Drug B. | 83 | Drug A may increase the vasopressor activities of Drug B. |
| 41 | The risk or severity of hyperkalemia can be increased when Drug A is combined with Drug B. | 84 | Drug A may increase the tachycardic activities of Drug B. |
| 42 | The protein binding of Drug B can be decreased when combined with Drug A. | 85 | The risk of a hypersensitivity reaction to Drug B is increased when it is combined with Drug A. |

Table 9: List of 86 DDI types and their corresponding IDs from the DrugBank dataset. Generic placeholders "Drug A" and "Drug B" are used to denote the interacting pair.

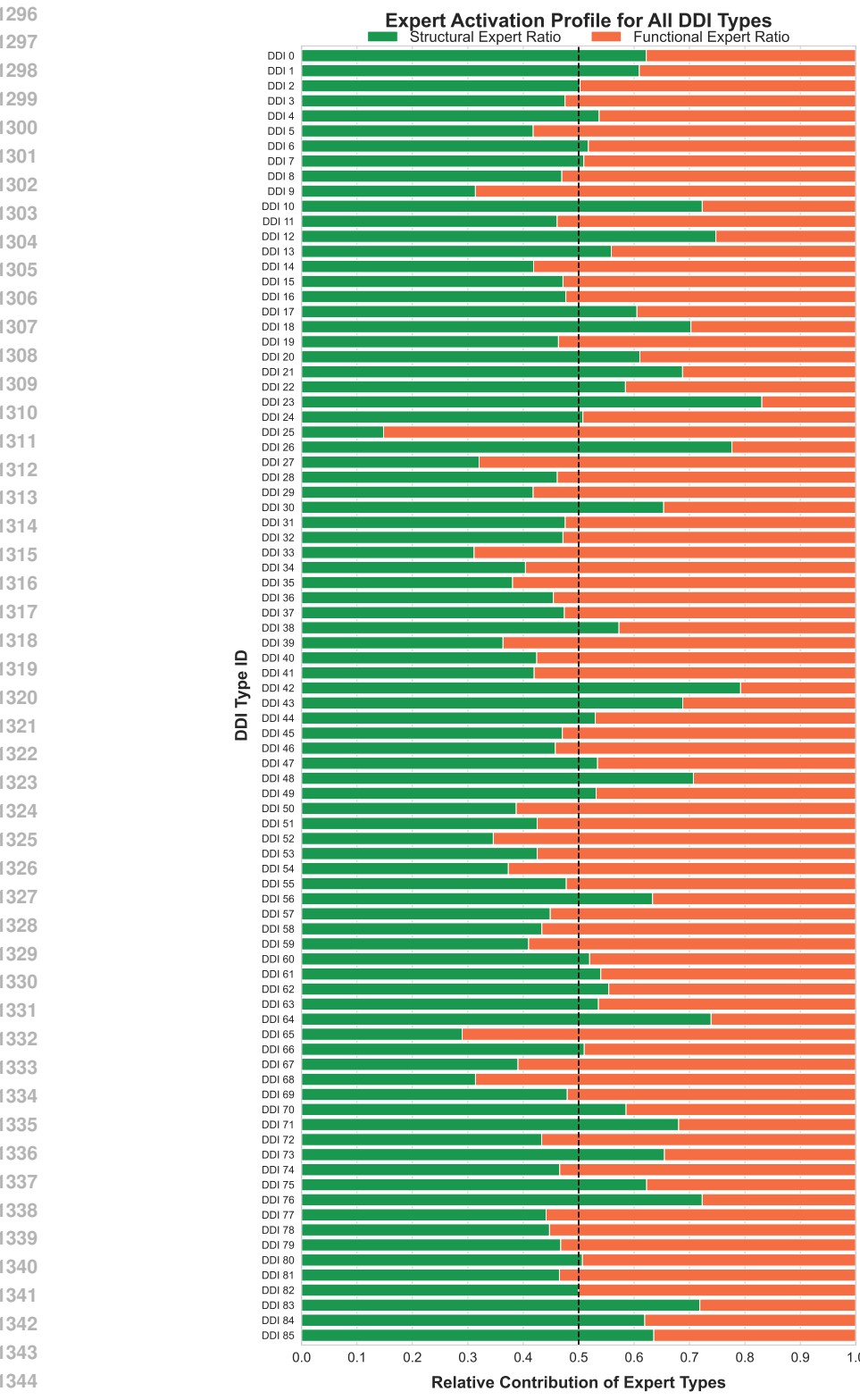

Figure 9: Comprehensive expert activation profiles for a diverse set of DDI types with established pharmacokinetic (PK) or pharmacodynamic (PD) dominance.

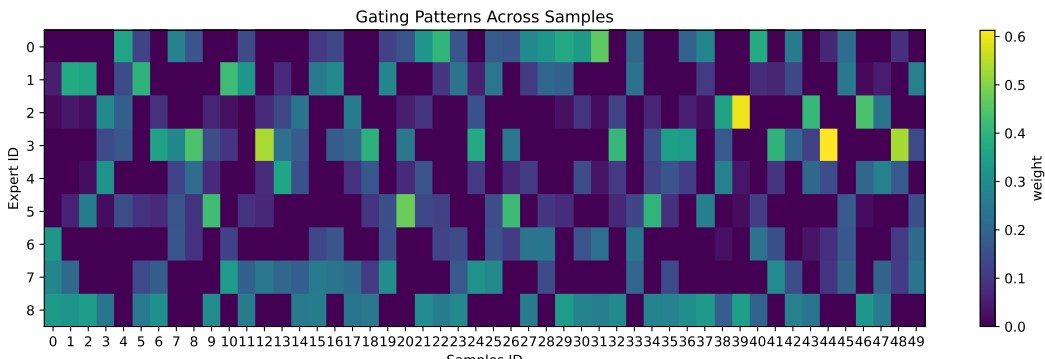

Figure 10: Visualization of expert-selection dynamics on 50 randomly sampled drug pairs from the DrugBank S1 split. Each column corresponds to a drug pair and each row to one of the experts. Expert IDs 0–3 denote structural experts, IDs 4–7 denote functional experts, and ID 8 denotes the relational expert.

## L  ADDITIONAL ANALYSIS OF EXPERT SELECTION DYNAMICS

To further characterize the dynamics of expert selection, Figure 10 visualizes the normalized gating weights over all experts for 50 randomly sampled drug pairs from the DrugBank S1 split. Each column in the heatmap corresponds to a drug pair and each row to an expert, revealing several salient patterns. First, the set of selected experts varies substantially across drug pairs: some pairs are dominated by structural experts (e.g., sample 39 and 44), others by functional (e.g., sample 0 and 20) or relational experts (e.g., sample 2 and 35), and across the random samples every expert attains high activation for at least some inputs. This indicates that the router does not collapse onto a small, fixed subset of experts but instead adaptively explores the full expert pool. Second, within the selected experts, the gating scores are typically peaked rather than flat. For some drug pairs, one or two experts receive clearly dominant weights, whereas for others the weights are more evenly distributed among the selected experts. These observations demonstrate that $\text{M}^2\text{DDI}$ performs genuinely input-dependent routing, with sample-specific and mechanism-aware specialization patterns that are consistent with the case-study analysis presented in the Section 4.5.

## M  ROBUSTNESS TO ATC HIERARCHY DEPTH AND INCOMPLETENESS

In $\text{M}^2\text{DDI}$, ATC information is used only in the prior path of the dual-path gate, while the experts themselves always operate on structural, functional, and relational representations. ATC therefore serves as an auxiliary routing signal rather than a hard requirement for the model to function. In this section, we quantitatively study (i) the sensitivity of $\text{M}^2\text{DDI}$ to the depth of the ATC hierarchy and (ii) its robustness to incomplete ATC coverage, with a particular focus on the implications for rare drugs.

**Sensitivity to ATC hierarchy depth.**   To assess how much $\text{M}^2\text{DDI}$ depends on the depth of ATC hierarchies, we vary the maximum ATC level used in the prior path on DrugBank, from only level-1 codes up to level-4 codes, and evaluate performance on the S1 and S2 test sets. All other components of the model and training procedure are kept fixed. The results are summarized in Table 10.

We observe that incorporating more detailed ATC information from level-1 to level-3 consistently improves performance on both S1 and S2. However, including level-4 categories does not yield further gains and even slightly degrades performance. This suggests that mid-level ATC information (levels 2–3) already captures the most useful pharmacological and therapeutic distinctions for expert routing, while overly fine-grained categories tend to introduce noise. Importantly, these results indicate that $\text{M}^2\text{DDI}$ does not require deep or fully specified ATC hierarchies: relatively coarse annotations (e.g., up to level-3) are sufficient to obtain most of the benefit. This property is particularly relevant for rare or newly approved drugs, for which only shallow ATC labels are often available.

Table 10: Effect of ATC hierarchy depth on $\mathtt{M^2DDI}$ performance (F1 score) on DrugBank (S1 / S2).

| ATC level | S1 | S2 |
|---|---|---|
| 1 | 67.14 | 38.76 |
| 2 | 67.85 | 39.64 |
| 3 | **68.28** | **40.52** |
| 4 | 67.83 | 39.25 |

Table 11: Robustness of $\mathtt{M^2DDI}$ to missing ATC annotations on DrugBank S1. We randomly mask ATC labels for a given fraction of drugs and keep all other modalities unchanged.

| Dataset | DrugBank | | | | | |
|---|---|---|---|---|---|---|
| Percentage | 0% | 10% | 20% | 30% | 40% | 50% |
| F1 | **68.28** | 68.16 | 68.02 | 67.58 | 67.25 | 66.92 |

**Robustness to incomplete ATC coverage.** We further study robustness to missing ATC annotations by randomly masking ATC codes for different proportions of drugs on DrugBank S1, while keeping all other modalities unchanged. Concretely, for a masking ratio $p$, we uniformly sample $p\%$ of drugs and remove their ATC information entirely from the prior path, leaving their structural and functional features intact. The results are reported in Table 11.

As the percentage of drugs with missing ATC increases (up to 50%), performance degrades only gradually, and the model remains strong overall. This behavior matches the intended design of the dual-path gate: when ATC information is missing or sparse—as is common for rare or newly approved drugs—the routing naturally falls back on the feature-query path based on structural and functional representations.

Overall, these results demonstrate that (i) $\mathtt{M^2DDI}$ is not overly sensitive to the depth of ATC hierarchies, and (ii) the model is robust to substantial ATC incompleteness. Together, they support the claim that our method can effectively handle rare drugs with shallow or partially specified ATC annotations.

## N  VISUALIZING EXPERT RELEVANCE UNDER DATA INCOMPLETENESS

To validate the interpretability of our gating mechanism, we analyze the evolution of expert weights under the data incompleteness scenario described in Section 4.4, where we progressively mask the functional and relational modalities while preserving the structural modality. As illustrated in Figure 11, the model exhibits a highly adaptive routing behavior as the missing rate increases from 0% to 50%. Specifically, we observe a consistent suppression of experts dependent on the increasingly unavailable functional and relational data. Crucially, the gating network compensates for this information loss by dynamically amplifying the weights of the Structural Experts (Experts 0–3), which process the intact molecular graphs. This significant shift in attention—redirecting focus from corrupted to reliable modalities—confirms that the $\mathtt{M^2DDI}$ framework does not merely fuse features statically, but intelligently identifies and prioritizes the most pertinent available evidence to ensure robust predictions.

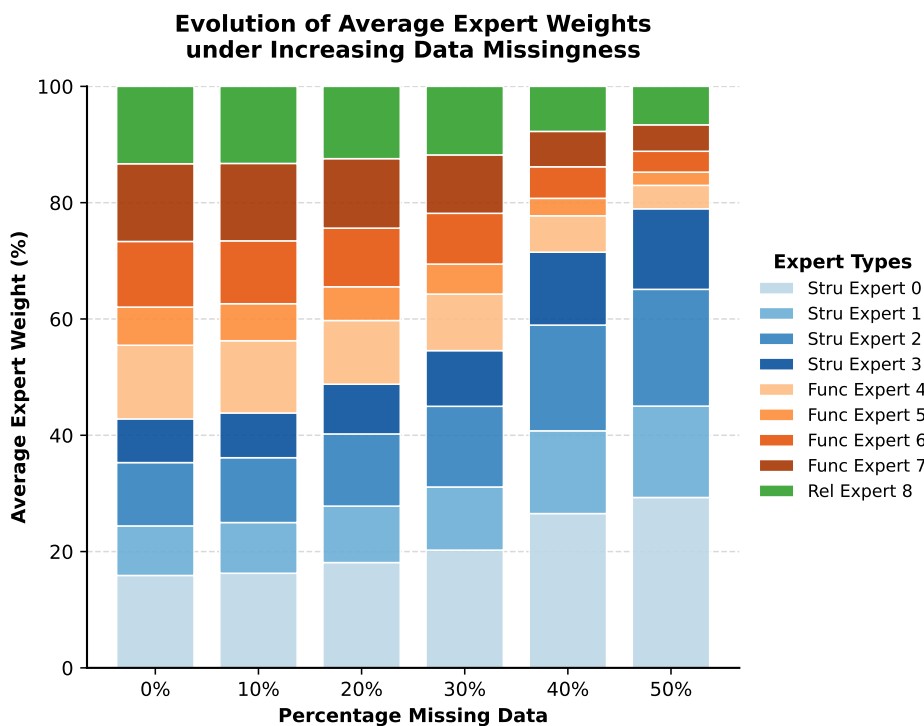

Figure 11: **Evolution of Average Expert Weights under Increasing Data Missingness.** The stacked bar chart displays the distribution of expert utilization as the percentage of missing functional and relational data increases (from 0% to 50%).

