# OpenReview forum: "M$^2$DDI: A Unified Framework for Dynamic Multimodal Fusion in Drug-Drug Interaction Prediction"
_ICLR.cc/2026/Conference — Submitted to ICLR 2026_

### Official Review · Reviewer_JFM8 · 2025-10-29

**Soundness:** 3
**Presentation:** 2
**Contribution:** 2
**Rating:** 4
**Confidence:** 3

**Summary:**

To jointly integrate molecular structure, pharmacodynamic function, and network-mediated relations in DDI prediction, this paper utilizes a Mixture-of-Experts (MoE) architecture, with each expert dedicated to a distinct pharmacological modality. A novel prior-enhanced dual-path gating strategy is proposed to adaptively select relevant experts for each drug pair by integrating mechanism-matched feature queries and Anatomical Therapeutic Chemical (ATC)-based biomedical priors, thereby aligning expert selection with underlying pharmacological mechanisms and addressing the challenge of data incompleteness. The proposed M^2DDI method was compared with seven state-of-the-art DDI methods on two widely used DDI datasets DrugBank and TWOSIDES.

**Strengths:**

S1: This work is to develop a unified framework that integrates different modalities to enhance the DDI prediction, outperforming single modality-based approaches.
S2: To address modality heterogeneity, dynamic relevance, and data incompleteness, a new multi-modal mixture-of-experts DDI prediction method is proposed.
S3: A novel dual-path gating strategy is embedded in M^2DDI by combining mechanism-matched feature queries and biomedical prior for dynamic expert selection.

**Weaknesses:**

W1: How to incorporate biomedical knowledge prior to guiding expert selection when data are incomplete or missing is not well elaborated. If the modality incompleteness is a major obstacle, this work should compare the proposed strategy with state-of-the-art solutions, which was mostly neglected by the current version.
W2: What is the difference and connection between the proposed M^2DDI and multi-head fusion attention strategy? The proposed strategy is to synergize a heterogeneous expert pool and a prior-enhanced dual-path gating strategy for expert selection, which is similar to multi-head fusion attention strategy.
W3: A prior-enhanced dual-path gating mechanism is designed to activate the top-K most relevant experts from a heterogeneous pool, and to dynamically decide which experts are relevant for a given input and how to weight their contributions. In the experiments, four structural experts, four functional experts, and one relational expert were configured while K was set-up to five for each prediction. Obviously, the number of relational experts insufficient and the rationale to determine K=5 is not discussed.

**Questions:**

- The motivation of addressing modality incompleteness or missing is lacking. Incorporating the discussion of dynamic relevance and data incompleteness regarding background and challenges in DDI would improve the paper.
- What is the difference and connection between the proposed M^2DDI and multi-head fusion attention strategy? The proposed strategy is to synergize a heterogeneous expert pool and a prior-enhanced dual-path gating strategy for expert selection, which is similar to multi-head cross-attention mechanism where each head could be considered an expert attending to different aspects of the drug pair.
- A prior-enhanced dual-path gating mechanism is designed to activate the top-K most relevant experts from a heterogeneous pool, and to dynamically decide which experts are relevant for a given input and how to weight their contributions. In the experiments, four structural experts, four functional experts, and one relational expert were configured while K was set-up to five for each prediction. Obviously, the number of relational experts insufficient and the rationale to determine K=5 is not discussed. Was this value empirically validated? An ablation study showing performance vs. different K values is essential.
- The paper claims to align expert selection with underlying pharmacological mechanisms. How are these mechanisms formally defined and mapped to the proposed modalities (structural, functional, relational)? Is there a many-to-many relationship between mechanisms and modalities?
- This paper would be strengthed by comparing graph-based fusion strategies instead of early vs. late fusion. The claim of dynamic multimodal fusion needs a more precise definition in the context of prior work.
- What are the two distinct paths in the gating network? Is one path processing the raw drug-pair features and the other processing the ATC-code priors? How are their outputs combined (e.g., additive, multiplicative, concatenated) to produce the final gating weights? How is the ATC-based prior knowledge represented and integrated? Is it a simple embedding lookup, or is there a more complex encoding process?
- Does the performance of M^2DDI degrade gracefully, and is the gating network's behavior interpretable when a key modality is missing (e.g., does it successfully re-weight its reliance to the available experts)?
- The M^2DDI method proposed in this paper overlaps to some extent with the Multimodal Feature Optimal Fusion Drug Interaction Prediction (MOF-DDI) method [1], which integrates features from multiple data sources to predict drug interactions. Specifically, the MOF-DDI model comprehensively considers literature descriptions of drug interactions, biomedical knowledge graphs, and drug molecular structures to predict drug-drug interactions.

[1] Wen, Q., Li, J., Zhang, C. and Ye, Y., 2023, October. A multi-modality framework for drug-drug interaction prediction by harnessing multi-source data. In Proceedings of the 32nd ACM International Conference on Information and Knowledge Management (pp. 2696-2705).

Minor errors:
In page 2, Prior-Enhanced Dual-Path Gating: M2DDI employs a Mixture-of-Experts architecture that includes a novel dual-path gating strategy, combining mechanism-matched feature queries and biomedical priors for dynamic expert selection, directly addressing
https://github.com/mechanism heterogeneity and data incompleteness. "https://github.com/mechanism" should be removed.

**Details Of Ethics Concerns:**

The validation of M^2DDI in new drug scenarios needs to be conducted carefully and comprehensively before applying it to real applications.

---

> ### Author Response · Authors · 2025-11-23
> **Responses to Reviewer JFM8 (1/4)**
>
> We sincerely thank the reviewer for the insightful feedback and for recognizing the value of our dual-path gating strategy in handling modality heterogeneity. In the following response, we clarify questions and comments raised by the reviewer.
>
> > **W1. How to incorporate biomedical knowledge prior to guiding expert selection when data are incomplete or missing is not well elaborated.**
>
> We thank the reviewer for highlighting this. We acknowledge that the strategy for missing priors was not explicitly detailed. We revised Section 3.3.2 to specify that the model employs a **learnable 'missing embedding'** to represent the absence of ATC codes, ensuring the gating mechanism remains operational.
>
> While ATC codes are standard for most approved drugs, our Dual-Path Gating is explicitly designed to handle cases where this prior knowledge is incomplete or missing:
>
> 1. **Architectural Resilience:** The gating score is derived from the integration of the Feature-Query Path ($S_{feat}$) and the Prior Path ($S_{ATC}$). These paths operate in parallel. If biomedical priors (ATC) are missing for a new drug, the Prior Path effectively contributes a neutral signal (or zero embedding), causing the gating network to naturally fall back to the Feature-Query Path. This allows the router to select experts solely based on available structural or functional data without breaking the inference pipeline.
> 2. **Empirical Verification**: We explicitly validated this scenario in our "Robustness to Incomplete ATC Coverage" experiment (Appendix M and Table 10). Results show that even when we randomly mask ATC codes for 50% of the drugs, the F1 score drops only marginally (from 68.28% to 66.92%). This confirms that while priors enhance performance, the model is not strictly dependent on them and functions robustly when knowledge data is incomplete.
>
> > **Q1: The motivation of addressing modality incompleteness or missing is lacking. Incorporating the discussion of dynamic relevance and data incompleteness regarding background and challenges in DDI would improve the paper.**
>
> We thank the reviewer for this suggestion. We have revised the Introduction to explicitly ground our motivation in two operational realities of drug discovery:
>
> 1. **Modality Incompleteness (The "Asymmetry" of Novel Drugs):** Real-world deployment faces inherent data asymmetry. For newly synthesized compounds, **Molecular Structure** is always available, whereas **KG connections** and **Textual Descriptions** are often sparse or missing (the "cold-start" problem). As noted by Gangwal et al. (2024), this scarcity causes conventional models that assume complete inputs to fail. Our framework is explicitly motivated to ensure resilience against this realistic "missingness."
> 2. **Dynamic Relevance (Mechanism Heterogeneity):** DDI mechanisms are heterogeneous: **Pharmacokinetic (PK)** interactions (e.g., metabolism) are structure-dependent, while **Pharmacodynamic (PD)** interactions (e.g., antagonism) rely on functional targets. Static fusion indiscriminately combines these, allowing noise from irrelevant modalities to dilute the critical signal. Our goal is to mimic pharmacological reasoning by adaptively weighting the specific modality that governs the interaction mechanism for each unique drug pair.
>
> > **Q2: What is the difference and connection between the proposed M²DDI and multi-head fusion attention?**
>
> We thank the reviewer for this insightful question. There are key architectural and functional differences between these two architecture:
>
> - **Heterogeneous experts vs. homogeneous heads.** In M²DDI, each “head” is a full expert network tied to a specific pharmacological view (PK/structural, PD/functional, or relational/KG), i.e., we route over mechanism- and modality-specific subnetworks. In contrast, multi-head fusion attention uses homogeneous heads that are different projections within the same feature space, without explicit alignment to PK/PD/relational mechanisms.
> - **Sparse routing vs. dense mixing.** Our gate performs Top-K sparse expert routing: only a small subset of experts is activated per drug pair, which improves interpretability and capacity scaling. Standard multi-head fusion attention typically aggregates all heads in a dense manner, so every head participates in every forward pass.
> - **Routing over modalities/mechanisms vs. feature-level mixing.** Our gating operates at the level of experts bound to specific modalities and mechanisms. Multi-head fusion attention is usually applied to token/feature sequences, mixing features within or across modalities, but not routing among distinct mechanism-specific subnetworks.
>
> In summary, M²DDI is better viewed as a sparse mixture-of-experts over mechanism-aligned subnetworks, rather than a re-branding of standard multi-head fusion attention.
>
> **Reference:**
>
> Gangwal, Amit, et al. "Current strategies to address data scarcity in artificial intelligence-based drug discovery: A comprehensive review."

---

> ### Author Response · Authors · 2025-11-23
> **Responses to Reviewer JFM8 (2/4)**
>
> > **Q3: Number of relational experts insufficient and the rationale to determine K=5 is not discussed.**
>
> Thank you for the question. As detailed in **Appendix H.1** and visualized in **Figure 6**, where we restate and directly address this concern, the optimal configuration $(N_r = 1, N_s + N_f = 8, K = 5)$ is selected based on the grid search over $(N_s\in [1,..., 7], N_f\in [1,..., 7], N_r\in [1,2,3], K\in [2,..., 7])$. Overall, this configuration provides a good trade-off between (i) **sufficient diversity and capacity**, (ii) **balanced representation across modalities with different parameter scales**, and (iii) **stable and efficient training of the sparse MoE gating mechanism**.
>
> > **Q4: The paper claims to align expert selection with underlying pharmacological mechanisms. How are these mechanisms formally defined and mapped to the proposed modalities (structural, functional, relational)? Is there a many-to-many relationship between mechanisms and modalities?**
>
> We thank the reviewer for raising this point. Our notion of “mechanism-aligned expert selection” is grounded in the **standard pharmacological distinction** between pharmacokinetic (PK) and pharmacodynamic (PD) drug–drug interactions, plus a third, network-level relational view.
>
> **(1) Formal definition of mechanisms and mapping to modalities.**
>
> In M²DDI, we map these to modalities as follows:
>
> - The **structural expert group** primarily captures PK-related properties (e.g., metabolism, transport, permeability) from the **molecular graph**, which is known to be strongly associated with ADME behavior.
> - The **functional expert group** captures PD-related information from **textual / functional descriptors**, including indications, targets, and mechanism descriptions.
> - The **relational expert** operates on the **KG subgraph**, encoding higher-order relational mechanisms (e.g., shared targets, pathways, or diseases) that complement PK/PD.
>
> The mapping itself is **not hard-coded in the loss**; instead, we design the expert pool and inputs so that each expert is *most suitable* for a particular mechanism type, and the gating network learns to prefer them when that mechanism is relevant.
>
> **(2) Many-to-many relationships and soft alignment.**
>
> We do **not** assume a one-to-one mapping between mechanisms and modalities. In practice, a single DDI can involve both PK and PD components, and PK/PD evidence may appear in multiple modalities (e.g., PK described in text, PD reflected in structure via known SAR). Our gating is **soft and many-to-many**: each drug pair receives a normalized weight distribution over all experts, and different modalities can be activated simultaneously.
>
> What we mean by “alignment” is that, **on average**, the learned routing correlates with pharmacological mechanism categories. In Section **4.5**  and Figure 5, we show that DDI types manually labeled as PK-dominant (based on DrugBank mechanism fields and clinical references) are routed with higher weights to structural experts, while PD-dominant types are routed more strongly to functional experts, with the relational expert contributing across both. This provides empirical evidence that the gating network leverages the expert pool in a **mechanism-consistent but non-exclusive** way.

---

> ### Author Response · Authors · 2025-11-23
> **Responses to Reviewer JFM8 (3/4)**
>
> > **Q5: This paper would be strengthed by comparing graph-based fusion strategies instead of early vs. late fusion. The claim of dynamic multimodal fusion needs a more precise definition in the context of prior work.**
>
> We thank the reviewer for this insightful comment.
>
> **(1) Rationale for MoE over Graph-based Fusion**
>
> We prioritize the Mixture-of-Experts (MoE) architecture over graph-based fusion based on both empirical evidence and domain suitability:
>
> - **Literature Evidence:** Recent studies (e.g., REMOTE [1]) demonstrate that MoE systematically outperforms graph-based fusion. Graph-based methods often suffer from **"over-smoothing,"** where distinct signals (e.g., fine-grained textual semantics) are diluted during global message passing.
> - **Mechanism Suitability:** Structural (PK) and Textual (PD) features represent intrinsic properties independent of graph topology. Forcing these into a unified graph fusion risks obscuring precise molecular signals with noisy neighborhood information. By treating the KG as just one expert among many, our MoE architecture preserves the semantic integrity of intrinsic modalities, fusing them only when the gating mechanism determines relevance.
>
> **(2) Definition of "Dynamic Multimodal Fusion"**
>
> We clarify that our usage of "dynamic" aligns with modern multimodal literature, defined as input-dependent, sample-wise adaptability:
>
> - **Static/Global Fusion:** Graph-based or concatenation methods employ a "fixed computation graph" where all samples undergo the same aggregation, regardless of whether the interaction is structurally or functionally driven.
> - **Dynamic/Instance-wise Fusion (M²DDI):** Our framework performs **active routing**, generating a specific expert distribution for each drug pair (e.g., activating only Structural + Textual experts while ignoring Relational ones). This sample-wise architectural reconfiguration constitutes "dynamic fusion."
>
> **Reference:**
>
> [1] Lin, Xinkui, et al. "REMOTE: A Unified Multimodal Relation Extraction Framework with Multilevel Optimal Transport and Mixture-of-Experts." *Proceedings of the 33rd ACM International Conference on Multimedia*. 2025.
>
> > **Q6: Could the authors clarify the gating network design?**
>
> We wish to clarify that the dual-path gating mechanism is specified in **Sec. 3.3.2 (Eq. (1)–(5))** and **Figure 2**. To ensure clarity, we briefly recap the design here:
>
> - **What are the two paths?**
>   -  The gating network has two parallel scoring paths (Section 3.3.2, Eq. (1)–(5)):
>   -  **Path 1 – feature-query path (raw multimodal features).** We encode each drug’s molecular graph and textual description, concatenate the resulting structural and functional representations into a single drug-pair vector, and feed it into a learnable expert-scoring module to obtain feature-based expert scores (Eq. (1)). Intuitively, this path asks: *“Given the actual structural and functional features of this drug pair, which experts should be activated?”*
>   -  **Path 2 – prior path (ATC-based priors only).** Independently of the raw features, we construct ATC-based representations by embedding the hierarchical ATC codes, aggregating them at the drug level, and forming a drug-pair ATC vector, which is then mapped to a second expert score vector (Eq. (2)–(4)). This path provides expert scores purely from pharmacological priors, answering: *“Based only on ATC-derived pharmacological priors, which experts should be activated?”*
> - **How are the two paths combined?**
>   -  We first compute both score vectors $s_{\text{feat}}$ and $s_{\text{ATC}}$ independently (Eq. (1), Eq. (4)). We then **concatenate** them and pass the result through a routing matrix $W_{\text{gate}}$ followed by a sigmoid activation (Eq. (5)).
>   -  The resulting vector $\mathbf{g}$ contains one gating score per expert. In Sec. 3.4 (Eq. (6)–(8)), we use these scores to select the **Top-K experts** and then apply a softmax over their scores to obtain the final **mixture weights** used in the weighted sum of expert outputs.
> - **Is ATC just a simple lookup?**
>   -  No. ATC information is **not** fed as a single ID embedding. It is encoded through a **hierarchical, multi-level representation**: each ATC code is broken into levels, each level has its own embedding, these are concatenated and linearly projected, and then multiple codes per drug are aggregated by mean pooling before being mapped into expert scores. This multi-step encoding (Eq. (2)–(4)) is designed to respect the hierarchical structure of ATC and to let the model learn which ATC levels are most informative for gating.

---

> ### Author Response · Authors · 2025-11-23
> **Responses to Reviewer JFM8 (4/4)**
>
> > **Q7: Does the performance of M²DDI degrade gracefully, and is the gating network’s behavior interpretable when a key modality is missing?**
>
> We appreciate the reviewer’s question about behavior under missing modalities.
>
> **Graceful degradation of performance.**
>
> We explicitly study modality incompleteness in our robustness experiments. In Section 4.4, we progressively remove the **functional** and **relational** modalities while always keeping the structural view, and compare M²DDI with both multimodal fusion baselines (Static Fusion, Ensemble) and strong graph-based DDI models. As shown in Figure 4, across all dropout levels, the performance of M²DDI **decreases smoothly rather than collapsing**, and it **consistently maintains the best macro-F1 / AUPR** among all methods. These results indicate that the model **does degrade gracefully** and can still leverage the remaining modalities effectively when one source is missing.
>
> **Interpretable re-weighting of experts under missing modalities.**
>
> We analyzed the evolution of expert weights under the robustness setting (same as section 4.4), where **functional** and **relational** modalities were progressively masked while the structural modality was preserved. As shown in the **Figure 10**, as the missing rate increases from 0% to 50%, the gating network exhibits a clear trend: it automatically suppresses experts dependent on the missing modalities and compensates by significantly amplifying the weights of **Structural Experts** (which process the intact **Molecular Graphs**, e.g., Experts 0–3). This adaptive shift confirms that the gating mechanism interpretably distinguishes between corrupted and reliable signals, dynamically pivoting attention to the intact structural evidence to mitigate data incompleteness.  We include this visualization and analysis in **Appendix N** of the revision.
>
> > **W1: If the modality incompleteness is a major obstacle, this work should compare the proposed strategy with state-of-the-art solutions, which was mostly neglected by the current version.**
> >
> > **Q8: M²DDI overlaps to some extent with the MOF-DDI**
>
> We thank the reviewer for highlighting MOF-DDI. While we acknowledge it as a relevant multimodal work, M²DDI distinguishes itself through a fundamental paradigm shift in how heterogeneity and incompleteness are handled: **Static Feature Alignment vs. Dynamic Expert Routing**.
>
> 1. **Methodological Divergence (The Core Difference):** MOF-DDI employs **Optimal Transport** to project heterogeneous embeddings into a unified latent space. This represents a **Static Alignment** paradigm: it implicitly assumes all modalities should be aligned and contribute to every prediction. This approach struggles with **Mechanism Heterogeneity**, where a DDI is driven *exclusively* by one modality (e.g., purely structural PK interactions) rather than a consensus of all features. In contrast, M²DDI treats modalities as independent **experts**. Instead of forcing alignment, we use **Dynamic Routing** to actively select the single most relevant evidence source for each drug pair. This allows our model to "switch" between pharmacological mechanisms (Structure vs. Text vs. KG) rather than averaging them.
> 2. **Robustness to Modality Incompleteness:** Regarding the reviewer's concern on incompleteness: MOF-DDI's alignment strategy is inherently fragile when a target modality is entirely missing (common for new drugs), as the optimal transport plan becomes ill-defined. M²DDI handles this natively: the gating mechanism simply **down-weights or ignores the expert** associated with the missing modality, routing the decision to available experts (e.g., using Structure when KG is absent), ensuring robust inference in S1/S2 settings.
>
> *Note: Due to the lack of publicly available code or preprocessed data for MOF-DDI, direct empirical reproduction is infeasible. We will, however, include this detailed theoretical comparison in the Related Work section.*

---

> ### Comment · Area_Chair_7Lv5 · 2025-11-28
> **Rebuttal Review Request**
>
> Dear Reviewers,
>
> Thank you for your time and thoughtful feedback on this manuscript.
>
> The authors have now submitted their rebuttal. If you haven’t already, we kindly ask you to review their responses and consider whether your concerns have been adequately addressed.
>
> Best regards,
>
> AC

---

### Official Review · Reviewer_pYSV · 2025-10-30

**Soundness:** 2
**Presentation:** 2
**Contribution:** 2
**Rating:** 2
**Confidence:** 4

**Summary:**

The paper accurately identifies a core challenge in DDI prediction: existing methods struggle to jointly model the heterogeneous pharmacological mechanisms underlying DDIs. To address this, the authors propose M2DDI, an innovative Mixture-of-Experts (MoE) framework. This framework assigns specialized experts to each pharmacological modality, offering a logical and elegant architectural solution to this fundamental problem.

**Strengths:**

The paper accurately identifies a core challenge in DDI prediction: existing methods struggle to jointly model the heterogeneous pharmacological mechanisms underlying DDIs. To address this, the authors propose M2DDI, an innovative Mixture-of-Experts (MoE) framework. This framework assigns specialized experts to each pharmacological modality, offering a logical and elegant architectural solution to this fundamental problem.

**Weaknesses:**

1.	The evaluation metrics are insufficient. In multi-class DDI studies, it is standard to report six core metrics: Accuracy (ACC), AUC, AUPR, F1, Recall, and Precision.
2.	The model's strong generalization for new drugs (S1/S2 scenarios) is a key advantage. The problem, however, is that this ability is heavily reliant on ATC prior knowledge. According to the experiments in Figure 7, the model's performance in S1 and S2 scenarios drops "markedly" without this ATC path. This indicates its generalization isn't truly learned but rather "borrowed" from an external knowledge base. This creates a risk: if faced with a brand-new compound that lacks an ATC code, the model's predictive performance could plummet.
3.	The baseline comparison is incomplete: the model is not evaluated against multimodal DDI methods published in 2024–2025.
4.	Within the multimodal fusion framework, the paper lacks cross-modal interpretability.
5.	In a realistic novel-drug setting, a new drug typically comes with only partial information. However, if “new drugs” are selected directly from an existing knowledge graph, those attributes are often already fully recorded in the graph—so they are not truly novel or missing. This choice can overstate the information available to the model and deviate from real-world deployment conditions.

**Questions:**

1.	The evaluation metrics are insufficient. In multi-class DDI studies, it is standard to report six core metrics: Accuracy (ACC), AUC, AUPR, F1, Recall, and Precision.
2.	The model's strong generalization for new drugs (S1/S2 scenarios) is a key advantage. The problem, however, is that this ability is heavily reliant on ATC prior knowledge. According to the experiments in Figure 7, the model's performance in S1 and S2 scenarios drops "markedly" without this ATC path. This indicates its generalization isn't truly learned but rather "borrowed" from an external knowledge base. This creates a risk: if faced with a brand-new compound that lacks an ATC code, the model's predictive performance could plummet.
3.	The baseline comparison is incomplete: the model is not evaluated against multimodal DDI methods published in 2024–2025.
4.	Within the multimodal fusion framework, the paper lacks cross-modal interpretability.
5.	In a realistic novel-drug setting, a new drug typically comes with only partial information. However, if “new drugs” are selected directly from an existing knowledge graph, those attributes are often already fully recorded in the graph—so they are not truly novel or missing. This choice can overstate the information available to the model and deviate from real-world deployment conditions.

---

> ### Author Response · Authors · 2025-11-23
> **Responses to Reviewer pYSV (1/2)**
>
> > **Q1: The evaluation metrics are insufficient. You should use Accuracy (ACC), AUC, AUPR, F1, Recall, and Precision.**
>
> We thank the reviewer for the suggestion. While we acknowledge the value of Precision and Recall, there is no universally adopted “six-metric standard” in DDI literature. Instead, we follow the established dataset-dependent protocols used by SOTA baselines (e.g., SumGNN, TextDDI, EmerGNN) to ensure fair comparison. We justify our metric selection and the inclusion of additional metrics as follows:
>
> **1. DrugBank (Multi-class, Imbalanced):**
>
> - **F1-Score (Macro) over Precision/Recall:** We prioritize Macro F1-Score because it is the **harmonic mean of Precision and Recall**. It intrinsically captures the trade-off between these two measures, directly addressing the reviewer's interest in P/R while providing a single, robust scalar that prevents majority classes from dominating the evaluation.
> - **Cohen’s Kappa :** To further address the reviewer’s concern, we have added **Cohen’s Kappa** in Appendix Table 7. Unlike simple Accuracy, Cohen’s Kappa measures inter-rater agreement while **correcting for chance**. This is particularly critical for DrugBank's imbalanced distribution, as it rigorously discounts the possibility of the model achieving high scores merely by guessing the majority class.
>
> **2. TWOSIDES (Multi-label):**
>
> - We retain ROC-AUC and PR-AUC as they are the standard for multi-label ranking tasks. To be comprehensive, we have also added **Accuracy** in the Appendix.
>
> M²DDI demonstrates consistent superiority across these rigorous metrics, confirming that our performance gains are robust and not artifacts of metric selection.
>
> > **Q2: Model's strong generalization for new drugs is heavily reliant on ATC prior knowledge.**
>
> We appreciate the reviewer’s scrutiny regarding the role of ATC priors. We provide three lines of evidence to clarify that ATC acts strictly as a coarse routing prior to stabilize expert selection, rather than the primary driver of generalization:
>
> **1. Superiority without ATC:** As shown in the ablation study (Figure 3), even when the ATC path is removed entirely, M²DDI still achieves state-of-the-art performance across all settings (S0–S2). It continues to outperform all non-ATC baselines, including strong graph-based methods (e.g., EmerGNN, TIGER).
>
> **2. ATC alone is insufficient:** To verify that the model is not simply "borrowing" performance from external knowledge, we trained a baseline "ATC-only MLP" using only ATC representations. As shown in the table below, this baseline performs drastically worse than M²DDI (even the version without ATC).
>
> **Table: Performance of an ATC-only MLP baseline on DrugBank.**
>
> | **Method**         | **S1**    |         | **S2**      |           |
> | ------------------ | --------- | --------- | --------- | --------- |
> |                    | F1       | Acc        | F1       |Acc        |
> | ATC-only MLP       | 22.06     | 30.75     | 14.30     | 15.42     |
> | **M²DDI**(w/o ATC) | 64.43     | 70.79     | 37.13     | 45.57     |
> | **M²DDI**          | **68.28** | **71.73** | **40.52** | **46.49** |
>
> The substantial performance gap (e.g., 14.30 vs. 40.52 in S2 F1) confirms that strong inductive performance stems from the **multimodal expert architecture and dual-path gating**, not from the ATC features themselves.
>
> **3. Robustness**: As detailed in **Appendix M**, the model maintains high performance even when ATC hierarchies are shallow or when annotations are missing for 50% of drugs. This further proves that ATC is an auxiliary prior, not a hard requirement for generalization.

---

> ### Author Response · Authors · 2025-11-23
> **Responses to Reviewer pYSV (2/2)**
>
> > **Q3: The model is not evaluated against multimodal DDI methods published in 2024–2025.**
>
> We thank the reviewer for this suggestion. We respectfully clarify that **TIGER (AAAI 2024)** is already included as a primary baseline in our main experiments (Table 2). Beyond task settings, M²DDI distinguishes itself from TIGER through two fundamental methodological advancements:
>
> 1. **Fusion Paradigm: Static vs. Dynamic**
>
>     TIGER employs a **Static Fusion** paradigm, where molecular and knowledge graph features are fused via fixed concatenation and a shared MLP for all samples. This "one-size-fits-all" approach fails to account for the **heterogeneity of DDI mechanisms** (e.g., PK vs. PD). In contrast, M²DDI introduces **Dynamic Mixture-of-Experts Fusion**. Our gating mechanism performs **instance-wise routing**, adaptively prioritizing structural or relational experts based on the specific drug pair. This dynamic adaptability drives our performance gains over TIGER's static dual-channel architecture.
> 2. **Modality Compreteness: Graph-only vs. Text-Enhanced**
>
>     TIGER relies exclusively on **Molecular Graphs** and **Biomedical KGs**, overlooking unstructured **Textual Data** (e.g., mechanism-of-action). Structural and relational views often miss subtle pharmacodynamic details. M²DDI explicitly integrates a **Functional Expert (Text-based)**, providing complementary semantic signals that graph-only methods like TIGER inherently lack.
>
> > **Q4: Within the multimodal fusion framework, the paper lacks cross-modal interpretability.**
>
> We respectfully disagree with the claim that our multimodal fusion framework lacks cross-modal interpretability. In **Sec. 4.5 (“Case Study of Expert Specialization”) and Figure 5**, we already provide an explicit **cross-modal interpretability analysis** of the learned fusion behavior.
>
> Concretely, we curate a set of DrugBank S0 interactions with **established pharmacokinetic (PK)–driven vs. pharmacodynamic (PD)–driven mechanisms**, based on authoritative pharmacology literature, and then analyze the **relative contributions of structural vs. functional experts** for each DDI type. As shown in **Figure 5**, PK-driven interactions (e.g., metabolism-inhibition–driven DDIs) receive substantially higher weights from **structural experts**, whereas PD-driven interactions (e.g., target-synergistic or antagonistic DDIs) are predominantly routed to **functional experts**. This behavior demonstrates that the gating network learns to **prioritize different modalities in a mechanism-consistent way**, providing a clear, cross-modal explanation of how structural and textual evidence are used for different interaction types.
>
> > **Q5: Selecting “new drugs” directly from an existing knowledge graph makes them appear fully known, overstating available information and diverging from real-world novel-drug scenarios where data are incomplete.**
>
> We appreciate the reviewer's scrutiny regarding the inductive setting. We clarify that our experimental design strictly prevents the model from accessing "future" knowledge about new drugs, and our architecture is specifically built to handle the resulting data incompleteness:
>
> 1. **Strict Inductive Protocol (Random Split & Edge Removal):** Following previous works (SumGNN and EmerGNN), our S1/S2 splits are **randomly partitioned** into disjoint sets to ensure unbiased evaluation. To strictly simulate the "unknown" nature of novel drugs, we explicitly **remove all relational edges** associated with test drugs from the training Knowledge Graph. Consequently, during training, the model treats these test drugs as isolated nodes within the relational view, ensuring zero information leakage from the KG structure.
> 2. **Cross-Modal Compensation Mechanism:** Crucially, our framework addresses the challenge of these "isolated" new drugs through multimodal compensation. When a new drug lacks connections in the KG (rendering the Relational Expert less informative), M²DDI naturally **compensates by leveraging the Structural and Functional experts**. As demonstrated in our robustness analysis (Figure 4 and Figure 10 in Appendix N), the gating mechanism adaptively shifts its reliance to molecular structures and textual descriptions—data that is typically available even for brand-new compounds—thereby maintaining high predictive performance despite the incompleteness of the knowledge graph.

---

> ### Comment · Area_Chair_7Lv5 · 2025-11-28
> **Rebuttal Review Request**
>
> Dear Reviewers,
>
> Thank you for your time and thoughtful feedback on this manuscript.
>
> The authors have now submitted their rebuttal. If you haven’t already, we kindly ask you to review their responses and consider whether your concerns have been adequately addressed.
>
> Best regards,
>
> AC

---

### Official Review · Reviewer_3SEy · 2025-10-31

**Soundness:** 3
**Presentation:** 3
**Contribution:** 3
**Rating:** 6
**Confidence:** 3

**Summary:**

This paper proposes a unified multimodal framework for DDI prediction based on MoE architecture. Each expert corresponds to a distinct pharmacological modality and a dual-path gating mechanism dynamically selects the most relevant experts for each drug pair. The gating combines feature-based similarity and prior biomedical knowledge via ATC embeddings, aligning expert selection with underlying pharmacological mechanisms. Experiments show SOTA performance, particularly in new drug scenarios, and robustness against incomplete modality information.

**Strengths:**

* This paper presents a novel unified multimodal framework that fuses structural, functional, and relational modalities.

* The dual-path gating that integrates data-driven features with pharmacological priors is elegant and biologically meaningful.

* The $M^2DDI$ achieves excellent performance in unseen-drug scenarios.

* The alignment between expert activation and known pharmacological mechanisms provides a mechanistic understanding of DDI.

**Weaknesses:**

* The overall technical components are standard, and the innovation lies primarily in their combination rather than in new algorithmic formulations.

* The experiments lack direct comparison with recent multimodal foundation models.

**Questions:**

* How does this method perform compared to recent large pretrained multimodal models (e.g., DDI-GPT [1]) on the same datasets?

* What is the training/inference cost compared to early/late fusion models?

* How sensitive is the model to the depth or incompleteness of ATC hierarchies, especially for rare drugs?

[1] Xu, Chengqi, et al. "DDI-GPT: Explainable Prediction of Drug-Drug Interactions using Large Language Models enhanced with Knowledge Graphs." BioRxiv (2024): 2024-12.

---

> ### Author Response · Authors · 2025-11-23
> **Responses to Reviewer 3SEy (1/2)**
>
> We sincerely thank the reviewer for the positive assessment and the encouraging comments. We have carefully considered your comments and provide detailed responses to these points below.
>
> > **W1: The overall technical components are standard, and the innovation lies primarily in their combination rather than in new algorithmic formulations.**
>
> We appreciate the comment. While utilizing established primitives, our contribution is a **novel architectural paradigm** specifically designed to solve three challenges—**Modality Heterogeneity, Dynamic Relevance, and Data Incompleteness**—that standard fusion fails to address simultaneously:
>
> 1. **Methodological Innovation: A Unified Framework for Dynamic Evidence Reasoning.** Unlike static fusion, we restructure the prediction pipeline into a dynamic reasoning process:
>    - **Handling Heterogeneity & Relevance:** By architecturally decoupling evidence into specialized experts, we enable the model to process distinct data distributions independently. This allows it to dynamically **identify and prioritize** the most mechanism-relevant modality (e.g., Structural vs. Functional), preventing signal dilution.
>    - **Addressing Incompleteness:** We elevate dynamic fusion to an **availability-aware selection process**. In data-scarce scenarios (e.g., cold-start), the gating mechanism dynamically **re-calibrates expert activation** based on data presence. It actively shifts expert selection away from missing modalities to rely on available evidence channels (updated Figure 11 in Appendix N), ensuring the model adapts its routing topology rather than failing due to sparse inputs.
>    - **Significance:** This architecture offers a generalizable blueprint for biological tasks facing similar sparsity and complexity, such as **Drug-Target Interaction (DTI)**.
> 2. **Domain Innovation: Mechanism-Aware Pharmacological Modeling.** M²DDI moves beyond "black-box" combination to explicit pharmacological modeling. Recognizing that DDIs are driven by distinct mechanisms—**Pharmacokinetic (PK)** or **Pharmacodynamic (PD)**—we align experts with these pathways to achieve **dynamic mechanism matching**. This enables the model to adaptively switch its reasoning logic based on the specific interaction type, effectively solving the modality imbalance problem.
>
> > **W2 & Q1: The experiments lack direct comparison with recent multimodal foundation models (DDI-GPT).**
>
> We thank the reviewer for mentioning DDI-GPT. While relevant, DDI-GPT targets **Binary DDI Detection** (interaction vs. no interaction), whereas our work addresses **Fine-grained DDI Event Prediction** (predicting specific interaction types/side effects). A direct comparison is infeasible due to distinct output spaces. We emphasize that our setting is both technically more challenging and clinically more realistic:
>
> 1. **Higher Technical Complexity (The "Harder" Aspect):** Binary detection only requires identifying a generic "conflict signal." In contrast, our task involves predicting **86 distinct interaction types** (DrugBank) and **200 specific side effects** (TWOSIDES)(). This forces the model to **disentangle complex pharmacological mechanisms** (e.g., distinguishing pharmacokinetic metabolism inhibition from pharmacodynamic target antagonism) rather than simply flagging structural incompatibility.
> 2. **Superior Clinical Utility (The "Practical" Aspect):** In real-world Clinical Decision Support Systems (CDSS), binary alerts often lead to **alert fatigue** due to a lack of context. Our fine-grained setting provides **actionable insights**—for instance, distinguishing between "decreased efficacy" (requiring dosage adjustment) and "severe toxicity" (requiring discontinuation). This level of granularity is essential for actual medical decision-making, which binary models cannot support.

---

> ### Author Response · Authors · 2025-11-23
> **Responses to Reviewer 3SEy (2/2)**
>
> > **Q2:What is the training/inference cost compared to early/late fusion models?**
>
> We appreciate the reviewer’s question on computational cost. Appendix H (theoretical complexity) and Appendix I (empirical efficiency) already analyze the efficiency of M²DDI and show that its inference cost is asymptotically dominated by the relational expert. To directly address the comparison with early/late fusion, we additionally report early/late fusion models training time per epoch and worst-case inference time on DrugBank and TWOSIDES under identical settings:
>
> **Table: Efficiency comparison on DrugBank and TWOSIDES**
>
> | **Method**        | **DrugBank**   |                   |**TWOSIDES**    |       |
> | ----------------- | -------------- | ----------------- | -------------- | ----- |
> |  |Train/epoch (min)  |Inference (ms)  |Train/epoch (min)  |Inference (ms)       |
> | Static Fusion     | 29.42          | 59.04             | 4.33           | 68.03 |
> | Ensemble          | 30.03          | 59.16             | 4.46           | 68.12 |
> | M²DDI             | 11.8           | 59.35             | 1.5            | 68.25 |
>
> As shown in the table above, M²DDI trains approximately **2.5× faster** than fusion baselines while maintaining comparable inference latency. This significant speedup stems from the **conditional computation** inherent in our MoE architecture. Unlike Static or Ensemble methods that must execute the computationally intensive Relational Expert (GNN) for *every* sample to perform fusion, M²DDI only activates and updates the relational expert when selected by the gate. Since the relational GNN is the computational bottleneck (as detailed in Appendix I ()), bypassing it for a portion of samples drastically reduces the average training cost.
>
> > **Q3:How sensitive is the model to the depth or incompleteness of ATC hierarchies, especially for rare drugs?**
>
> We thank the reviewer for this important question. We emphasize that ATC information serves only as an auxiliary signal in the **prior path** of our dual-path gating mechanism. Since the experts themselves operate on structural, functional, and relational representations, the model does not treat ATC data as a hard requirement.
>
> To address the concern regarding rare drugs, we performed two robustness studies (added to **Appendix M**).
>
> **1. Sensitivity to Hierarchy Depth**
>
> We varied the maximum ATC level used in the prior path. As shown in the table below, performance peaks at Level 3. Adding Level 4 details introduces noise, while coarser annotations (Level 1-2) remain highly effective. This confirms M²DDI  does not require deep hierarchies, making it suitable for rare drugs with shallow labels.
>
> **Table: Effect of ATC hierarchy depth on M²DDI performance on DrugBank (S1/S2).**
>
> |               | **DrugBank** |           |
> | ------------- | ------------ | --------- |
> | **ATC Level** | S1           | S2        |
> | 1             | 67.14        | 38.76     |
> | 2             | 67.85        | 39.64     |
> | **3**         | **68.28**    | **40.52** |
> | 4             | 67.83        | 39.25     |
>
> **2. Robustness to Incompleteness**
>
> We simulated data sparsity by randomly masking ATC codes on DrugBank S1. As shown in the table below, the model exhibits strong resilience: even when **50% of drugs** lack ATC information entirely, the performance drop is marginal (< 1.4%).
>
> **Table: Robustness of M²DDI to missing ATC annotations on DrugBank S1.**
>
> |            | **DrugBank** |       |       |       |       |       |
> | ---------- | ------------ | ----- | ----- | ----- | ----- | ----- |
> | Percentage | 0%           | 10%   | 20%   | 30%   | 40%   | 50%   |
> | F1         | **68.28**    | 68.16 | 68.02 | 67.58 | 67.25 | 66.92 |
>
> **Mechanism of Robustness** This resilience is a direct result of the **dual-path gating strategy**. When ATC priors are missing or sparse (common for novel/rare drugs), the gating mechanism naturally adaptively falls back to the **feature-query path**, routing inputs based on the available structural and functional features(). We have included these results in **Appendix M** of the revised paper.

---

> ### Comment · Area_Chair_7Lv5 · 2025-11-28
> **Rebuttal Review Request**
>
> Dear Reviewers,
>
> Thank you for your time and thoughtful feedback on this manuscript.
>
> The authors have now submitted their rebuttal. If you haven’t already, we kindly ask you to review their responses and consider whether your concerns have been adequately addressed.
>
> Best regards,
>
> AC

---

### Official Review · Reviewer_URJh · 2025-10-31

**Soundness:** 2
**Presentation:** 3
**Contribution:** 2
**Rating:** 2
**Confidence:** 3

**Summary:**

This paper proposes M²DDI, a mixture-of-experts framework for drug-drug interaction prediction that combines molecular structure, textual information, and knowledge graph modalities through a dual-path gating mechanism.

**Strengths:**

This paper has a clear motivation, is well-written, and explores the impact of modality loss on DDI tasks.
- Clear problem formulation addressing modality heterogeneity, dynamic relevance, and data incompleteness
- Novel ATC-based prior path for handling missing modalities in new drugs

**Weaknesses:**

Although the architecture of this paper is well designed, it lacks novelty and theoretical justification. Moreover, issues such as insufficient experimental design and incomplete experiments weaken its contribution.

1. The paper compares Static Fusion and Ensemble as multimodal fusion approaches, but why not consider a more direct and computationally cheaper method? Specifically, obtaining embeddings from three modalities, learning a weight for each modality through an MLP, and then fusing them directly for prediction? This is important because M2DDI's improvement over Ensemble appears quite limited (S0: Ensemble 97.58 → 97.82; S2: Ensemble 46.45 → 46.49).

2. In the ablation study shown in Figure 3, can the authors provide the Accuracy results? From Table 2, the improvements in Accuracy are marginal (S0: EmerGNN 97.42, Ensemble 97.58 → 97.82; S2: EmerGNN 46.34, Ensemble 46.45 → 46.49).

3. Figure 4 aims to examine the impact of missing functional and relational modalities while always retaining structural information. However, why does the structure-based baseline "MLP" exhibit the steepest performance degradation? Moreover, it would be more convincing to include additional graph-based baselines in this robustness evaluation.

4. It remains unclear why the authors did not include more recent baselines, such as: [1] CARMEN: Context-Aware Safe Medication Recommendations with Molecular Graph and DDI Graph Embedding (AAAI 2023); [2] SSF-DDI: a deep learning method utilizing drug sequence and substructure features for drug-drug interaction prediction (2024); [3] DSN-DDI: an accurate and generalized framework for drug-drug interaction prediction by dual-view representation learning (2023); [4] Learning motif-based graphs for drug-drug interaction prediction via local-global self-attention.

5. Figure 6 demonstrates that the optimal configuration is Nr = 1, Ns + Nf = 8, K = 5, but can the authors provide an explanation for why this specific setting works best? More importantly, I am interested in understanding the expert selection dynamics during training. Are the scores of different experts often close to each other? Does the model always tend to select the same fixed set of k (5) experts, or does the selection vary significantly across different drug pairs? Can the authors clarify this point, perhaps by providing visualizations of expert utilization statistics or the distribution of gating scores across the dataset?

**Questions:**

See weaknesses.

---

> ### Author Response · Authors · 2025-11-23
> **Responses to Reviewer URJh (1/2)**
>
> We sincerely thank the reviewer for the constructive feedback and the time dedicated to evaluating our manuscript. We have conducted additional experiments and analyses to address them point-by-point below.
>
> > **Q1: Why not consider another computional cheaper fusion method as baseline?**
>
> We thank the reviewer for the suggestion. The proposed scheme—obtaining three modality embeddings, using an MLP to output modality weights, and then fusing them—is essentially a standard **early-fusion** design and a **special (less expressive) case** of our Static Fusion baseline. In Static Fusion, the first linear layer of the MLP already learns modality-specific and feature-wise weights, while the reviewer’s design restricts this to only three scalars. Computationally, this alternative is **not meaningfully cheaper**, as all fusion methods are dominated by the cost of running the three encoders (molecular GNN, BioMedBERT, KG GNN). **Importantly, M²DDI’s contribution goes far beyond a different static fusion head**: its heterogeneous experts, dual-path gating with ATC priors, and loss-free load balancing produce **sample-wise, mechanism-aware routing across modalities**, which a single static-fusion MLP cannot emulate.
>
> > **Q2: M²DDI's improvement over Ensemble appears quite limited (S0: Ensemble 97.58 → 97.82; S2: Ensemble 46.45 → 46.49)**
>
> Regarding the seemingly small Accuracy gains, we note that **Accuracy is not an informative metric** in DrugBank’s highly imbalanced **86-class** dataset, with an **imbalance ratio of 10,166**. As shown in table below, the DrugBank dataset exhibits a highly skewed, long-tailed label distribution. Notably, only **three DDI types** account for **62%** of all samples.  Following prior DDI works like SumGNN (**Bioinformatics**), TextDDI (**EMNLP**), and EmerGNN (**Nature Comp. Sci.**), we therefore use **macro-F1** (DrugBank) and **PR-AUC / ROC-AUC** (TWOSIDES), under which M²DDI shows **consistent and substantially larger improvements** than Static Fusion and Ensemble, especially in the **inductive new-drug (S1: Ensemble 61.31 → 68.28; S2: Ensemble 32.7 → 40.52) and modality-missing scenarios (Figure 4)** that are central to our contribution.
>
> **Table: Class frequency buckets in the DrugBank dataset**
> | **Frequency bucket (per class)** | **#** **Classes** | **% of classes** | **#** **Samples** | **% of samples** |
> | -------------------------------- | ----------------- | ---------------- | ----------------- | ---------------- |
> | **< 0.5%**                       | 68                | 79.1%            | 15057             | 7.8%             |
> | **0.5%–5%**                      | 15                | 17.4%            | 58091             | 30.2%            |
> | **>= 10%**                       | 3                 | 3.5%             | 119136            | 62.0%            |
>
> > **Q3: Can you provide the "Accuracy" results in the ablation study shown in Figure 3?**
>
> We appreciate the suggestion and have included the full Accuracy results in **Appendix H.3 and Figure 8**.
>
> As shown in Figure 8, our proposed M²DDI **consistently achieves the highest accuracy across all settings (S0, S1, and S2)** compared to the ablated variants. However, the performance gaps in Accuracy are narrower than those in Macro-F1 (Figure 3). This confirms our discussion in Q2: Accuracy is dominated by frequent classes, making it less sensitive to improvements. In contrast, Macro-F1 reveals the significant gains our model achieves on rare classes, which are the primary focus of this work.
>
> > **Q4: Why does the structure-only baseline “MLP” suffer the performance drop in Figure 4 despite always retaining structural information?**
>
> We thank the reviewer for the **insightful** comment and apologize that our original wording was not sufficiently clear. The phrase **“always-available molecular structures”** in Figure 4 refers specifically to the **fusion-based methods (static fusion / ensemble)**, where structural information is kept intact while functional and relational modalities are progressively removed. It does **not** imply that all baselines—including the single-modality **MLP**—should remain stable under this setting.
>
> Our robustness analysis is grounded in the realistic assumption that molecular graphs are essentially always available in practice. However, under this ablation protocol, a purely structural single-modality method like MLP does not align neatly with the design: if we do not reduce the amount of structural information it receives, its performance would remain nearly constant and become incomparable to models whose inputs are being progressively degraded; if we do reduce its only modality, we no longer reflect the “always-available structure” assumption. We agree that this is confusing, and in the revised version we therefore update the Figure 4 by removing the MLP curve and instead **including additional graph-based baselines** (TIGER and SumGNN) that are more consistent with this setting.

---

> ### Author Response · Authors · 2025-11-23
> **Responses to Reviewer URJh (2/2)**
>
> > **Q5: Why not add more baselines such as CARMEN, SSF-DDI, DSN-DDI, motif-based method?**
>
> We appreciate these references. However, our work focuses on the **comprehensive multi-class classification setting**, which is fundamentally distinct from and more challenging than the tasks addressed by the suggested methods.
>
> **Setting & Challenges:** M²DDI is designed to predict the **precise pharmacological interaction type** (e.g., distinguishing 86 specific types in DrugBank) for **unseen drugs** (Inductive S1/S2 settings). This imposes a high bar for model capability: it requires **fine-grained discrimination** of latent mechanisms rather than simple binary detection, and **robust feature generalization** rather than transductive graph completion.
>
> **Infeasibility of Suggested Baselines:** The suggested methods do not operate in this rigorous setting:
>
> - **SSF-DDI & DSN-DDI** are limited to **binary link prediction** (Interaction: Yes/No). They lack the architectural capacity to map drug pairs to high-dimensional interaction spaces without fundamental re-engineering.
> - **CARMEN** is a patient-centric recommendation system where DDIs serve only as auxiliary constraints, not as the prediction target.
> - **MeTDDI** is restricted to specific metabolism mechanisms and cannot generalize to the diverse interaction types evaluated in our work.
>
> Thus, we compared M²DDI against state-of-the-art baselines capable of handling this multi-class, inductive challenge (e.g., EmerGNN, TIGER).
>
> > **Q6: Can you provide an explanation for why this specific expert pool configuration works best?**
>
> Thank you for the question. As summarized in **Appendix H.1**, the optimal configuration $(N_r = 1, N_s + N_f = 8, K = 5)$ is selected based on the grid search over $(N_s\in [1,..., 7], N_f\in [1,..., 7], N_r\in [1,2,3], K\in [2,..., 7])$. Overall, this configuration provides a good trade-off between (i) **sufficient diversity and capacity**, (ii) **balanced representation across modalities with different parameter scales**, and (iii) **stable and efficient training of the sparse MoE gating mechanism**.
>
> > **Q7: Can you clarify expert selection dynamics during training?**
>
> Thanks for raising this question about the expert-selection dynamics. Our current submission already includes an analysis of the gating behavior in the main text (Section 4.5, Figure 5). There, we focus on clinically recognized DrugBank interactions that are known to be predominantly PK- or PD-driven. As described in the paper, Figure 5 shows that PK-driven DDIs receive higher weights on structural experts, while PD-driven DDIs are routed mainly to functional experts. This provides evidence that expert selection is input-dependent and aligned with underlying pharmacological mechanisms, rather than relying on a fixed subset of experts.
>
> To further address the variability and magnitude of expert scores, we have added a heatmap visualization of randomly sampled drug pairs in **Appendix L (Figure 9)**. This analysis confirms two key dynamics:
>
> 1. **Distinct Activation Patterns:** The subset of selected experts varies substantially across different drug pairs, confirming that the model does not rely on a fixed set of experts.
> 2. **Adaptive Weighting:** The gating weights are non-uniform and input-dependent, with the model dynamically assigning dominant weights to specific experts based on the sample context.
>
> Together, Figure 5 (mechanism-specific case study) and the new Figure 9 (random sample-wise gating heatmap) show that M²DDI performs genuinely input-dependent expert selection: gating scores are typically peaked rather than flat, and the chosen experts differ substantially across drug pairs. We incorporated this clarification and the additional visualization in **Appendix L**.

---

> ### Comment · Area_Chair_7Lv5 · 2025-11-28
> **Rebuttal Review Request**
>
> Dear Reviewers,
>
> Thank you for your time and thoughtful feedback on this manuscript.
>
> The authors have now submitted their rebuttal. If you haven’t already, we kindly ask you to review their responses and consider whether your concerns have been adequately addressed.
>
> Best regards,
>
> AC

---

### Author Response · Authors · 2025-12-02
**General Responses and Summary During This Special Period**

Dear Area Chair,

We sincerely appreciate your significant effort in stepping in to evaluate our work under these unprecedented circumstances. We understand that the lack of reviewer dialogue places a substantial responsibility on your independent assessment of our responses and the revised manuscript.

We are encouraged that the reviewers unanimously recognized the value and elegance of our proposed solution. Specifically, **Reviewer `pYSV`** praised the M²DDI framework as a "logical and elegant" solution; **Reviewer `JFM8`** explicitly highlighted the value of our "dual-path gating strategy" in effectively addressing modality heterogeneity; and **Reviewer `3SEy`** provided a positive assessment of the work's potential. This consensus affirms that M²DDI successfully establishes a new standard for fine-grained pharmacological reasoning.

To assist your evaluation in the absence of further reviewer dialogue, we have consolidated our rebuttal into a concise summary addressing the four primary areas of concern raised during the review process:

- **Core Contribution: Dynamic Reasoning vs. Static Fusion** We clarify that M²DDI introduces a paradigm shift from static feature mixing to mechanism-aware dynamic routing, explicitly solving **Modality Heterogeneity** and **Data Incompleteness** (`URJh` Q1, `3SEy` W1, `JFM8` W1/Q1/Q8). We empirically validate this dynamic capability: our new heatmaps **(Appendix L, Fig. 9)** and case studies **(Fig. 5)** confirm the resolution of *heterogeneity* via mechanism-aligned routing, while our robustness study (**Appendix N, Fig. 11**) proves the resolution of *incompleteness* via **adaptive compensation**—demonstrating that the gate automatically shifts reliance to available structural experts when relational data is missing. Furthermore, we confirm that this sparse conditional computation makes M²DDI train 2.5× faster than static/ensemble baselines (`3SEy` Q2).

- **Robustness & Generalization of ATC Priors** We decisively refute the concern that our performance relies on external ATC knowledge (`pYSV` Q2, `3SEy` Q3, `JFM8` W1). Our rigorous ablations **(Fig. 7)** show M²DDI achieves SOTA even when the ATC path is completely removed, whereas an ATC-only MLP baseline **(table in the response for `pySV` Q2)** fails (~14% F1 vs. our 40%). Moreover, our new "Robustness to Incompleteness" experiments **(Appendix M, Table 11)** confirm that randomly masking ATC codes for 50% of drugs results in negligible performance degradation (<1.4%), proving our Dual-Path Gating effectively falls back to feature-based routing when priors are missing.

- **Baseline Selection & Task Distinction** Regarding baseline selection, we emphasize that we target the technically rigorous Fine-grained Classification (86 types) setting, distinguishing our work from binary detection (e.g., DDI-GPT) or patient-centric systems (e.g., CARMEN) which lack the capacity for pharmacological reasoning (`URJh` Q4/Q5, `3SEy` W2/Q1). We benchmark against capable multimodal methods like TIGER and have expanded our Related Work to theoretically contrast with MOF-DDI (`JFM8` Q8), highlighting that our dynamic routing offers a superior architectural fit for handling missing modalities in inductive settings compared to optimal transport.

- **Clarifying Misconceptions on Metrics, Protocols, and Design** We addressed technical queries regarding evaluation and design (`URJh` Q1/Q2/Q6, `pYSV` Q1/Q5, `JFM8` Q3). We clarified that simple Accuracy is uninformative due to extreme class imbalance (>10,000 ratio) and added Cohen’s Kappa **(Appendix H, Table 7)** to validate our robust gains. We also reaffirmed that our strict inductive protocol—removing test edges—ensures zero information leakage, and clarified that the expert pool configuration was optimized via extensive grid search **(Appendix H.1)**.

We have strived to address every reviewer's inquiry with the utmost thoroughness in our revisions and detailed responses above. While we deeply regret that the system-wide issues precluded the opportunity for a deeper, interactive discussion with the reviewers, we remain confident that the solidity of our method and the comprehensiveness of our rebuttal stand firmly on their own. We sincerely hope that the value and contributions of our work will be recognized.

Sincerely,

The Authors

---

### Meta-Review · Area_Chair_jRYp · 2025-12-04

**Summary:**

The paper accurately identifies a core challenge in DDI prediction: existing methods struggle to jointly model the heterogeneous pharmacological mechanisms underlying DDIs. To address this, the authors propose M2DDI, an innovative Mixture-of-Experts (MoE) framework. This framework assigns specialized experts to each pharmacological modality, offering a logical and elegant architectural solution to this fundamental problem.

**Reviewer Concerns:**

Two reviewers clearly reject this paper. Most concerns are experimental validation. I do not think these responses have addressed reviewers' concerns.

**Reviewer Scores:**

No. I do not think reviewers will change their scores.

---

### Decision · Program_Chairs · 2026-01-26

Reject